# α-Weighted Federated Adversarial Training

## Abstract

Federated Adversarial Training (FAT) helps us address the data privacy and governance issues, meanwhile maintains the model robustness to the adversarial attack. However, the inner-maximization optimization of Adversarial Training can exacerbate the data heterogeneity among local clients, which triggers the pain points of Federated Learning. This makes that the straightforward combination of two paradigms shows the performance deterioration as observed in previous works. In this paper, we introduce an $\alpha$-Weighted Federated Adversarial Training ($\alpha$-WFAT) method to overcome this problem, which relaxes the inner-maximization of Adversarial Training into a lower bound friendly to Federated Learning. We present the theoretical analysis about this $\alpha$-weighted mechanism and its effect on the convergence of FAT. Empirically, the extensive experiments are conducted to comprehensively understand the characteristics of $\alpha$-WFAT, and the results on three benchmark datasets demonstrate $\alpha$-WFAT significantly outperforms FAT under different adversarial learning methods and federated optimization methods.

## 1 Introduction

To handle the data privacy and governance issues, Federated Learning (McMahan et al., 2017) as one promising paradigm of distributed training has drawn the increasing attention (McMahan et al., 2017; Kairouz et al., 2019). However, training locally in Federated Learning also introduces the vulnerability from the adversarial attacks (Goodfellow et al., 2015; Kurakin et al., 2016), which drives us to consider the model robustness in this framework. Thus, recent advances (Kairouz et al., 2019) explore to apply the Adversarial Training methods (Madry et al., 2018) into Federated Learning.

However, the straightforward combination of Adversarial Training and Federated Learning presents some potential challenges due to the communication cost and the hardware requirement. For example, Shah et al. (2021) pointed out that the communication in Federated Learning may be a constraint to Adversarial Training, and proposed a dynamic schedule on the local training epochs to achieve the expected robustness in a short communication budget. Hong et al. (2021) considered the hardware constraint where some clients are not able to participate Adversarial Training, and they proposed a federated robust propagation method to share the adversarial robustness among the clients. Although these previous works indeed addressed some realistic problems, one critical issue in the way is the performance deterioration in the combination of two paradigms as observed in (Zizzo et al., 2020).

As shown in the left panel of Figure 1, one typical phenomenon in Federated Adversarial Training (FAT) is the robust accuracy of FAT (Zizzo et al., 2020) based on FedAvg (McMahan et al., 2017) will decrease significantly at the later stage of learning compared with the centralized Adversarial Training (Madry et al., 2018) that does not. Actually, it exists in many variants of Federated Learning methods (Shah et al., 2021) under different communication rounds and different local training epochs. However, it is still lack of the sufficient algorithmic breakthrough to overcome this issue, since almost all previous works (Zizzo et al., 2020; Shah et al., 2021; Hong et al., 2021) consistently apply the conventional update framework of Federated Learning with Adversarial Training.

We dive into this phenomenon and attribute it to the inner-maximization of Adversarial Training. Compared with the centralized Adversarial Training (Madry et al., 2018), the training data of FAT is distributed to each client, which leads to the Adversarial Training in each client unaware of the data in the others. Therefore, the adversarial examples generated by the inner-maximization of Adversarial Training tend to be highly biased to each local distribution, yielding the radical optimization to pursuit the model robustness (as shown in Figure 2, it has a severe local bias to the global optimum). In

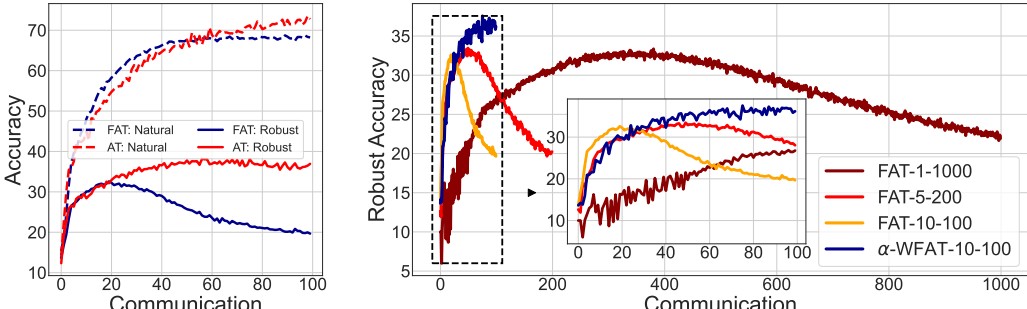

Figure 1: Left: comparison between centralized Adversarial Training and Federated Adversarial Training based on FedAvg. Right: comparison between FAT and $\alpha$-WFAT. All the experiments are conducted on *CIFAR-10* dataset (Non-IID) with 5 clients, and use AT (Madry et al., 2018) to train as well as PGD-20 to evaluate the Robust Accuracy. Note that, the notation "method-A-B" in the right panel means the method with A local training epochs and B communication rounds.

short, the inner-maximization of Adversarial Training exacerbates the data heterogeneity among local clients, which actually triggers the pain points of Federated Learning that are exploring urgently.

To handle this problem, we propose a new learning framework based on a simple but effective re-weighting mechanism, namely, $\alpha$-*Weighted Federated Adversarial Training* ($\alpha$-WFAT). Concretely, we relax the objective of the inner-maximization in Adversarial Training into a lower bound by an $\alpha$-weighting mechanism (as Eq. (3) in Section 4.2). Similar to the bias-variance trade-off (Kohavi et al., 1996), we introduce the small bias to the original objective via a low-bound relaxation, facilitating a friendly optimization in the combination of Adversarial Training and Federated Learning. This constructs a conservative optimization for Adversarial Training that weights different population based on their adversarial losses. Then, the harsh heterogeneous update can be down-weighted with this flexible weighting mechanism, and the convergence could be accelerated at the same time. Applying the similar idea to FAT, we propose $\alpha$-WFAT that emphasizes the robust clients more compared with other non-robust clients to alleviate the heterogeneous bias caused by the local adversarial generation. The right panel of Figure 1 gives a simple comparison between $\alpha$-WFAT and FAT. Empirically, we conducted extensive experiments to provide a comprehensive understanding of the proposed $\alpha$-WFAT, and the results of $\alpha$-WFAT in the context of different adversarial learning methods and federated optimization methods demonstrate its superiority to improve the model performance.

**Main Contributions**

- We derive an $\alpha$-weighted relaxation for Adversarial Training to relax the inner-maximization by a lower bound, which builds a mediating function to alleviate the potential radical optimization in its straightforward combination with Federated Learning (in Section 4.2).

- We propose a new learning framework, *i.e.,* $\alpha$-Weighted Federated Adversarial Training ($\alpha$-WFAT), to realize the relaxation of inner-maximization in FAT, which is simple and compatible with various Federated Learning or Adversarial Training methods (in Section 4.3).

- We conduct extensive experiments to comprehensively understand the characteristics of the $\alpha$-WFAT, and confirm its effectiveness on improving the model performance for both IID and Non-IID settings in the context of several federated optimization methods (in Section 5).

## 2 RELATED WORK

**Federated Learning**    The representative work in Federated Learning is FedAvg (McMahan et al., 2017), which has been proved effectiveness during the distributed training to maintain the data privacy. To further address the heterogeneous issues, several optimization approaches have been proposed *e.g.,* FedProx (Li et al., 2018), FedNova (Wang et al., 2020b) and Scaffold (Karimireddy et al., 2020). FedProx introduced a proximal term for FedAvg to constrain the model drift cause by heterogeneity; FedNova proposed a general framework that eliminated the objective inconsistency and preserved the fast convergence; Scaffold utilized the control variates to reduce the gradient variance in the

local updates and accelerate the convergence. MOON (Li et al., 2021a) alleviated the heterogeneity by maximizing the agreement between the representation of the local model and that of the global model, which helps correct the local training of individual parties. Reisizadeh et al. (2020) developed a robust federated learning algorithm to against distribution shifts in clients samples. Our $\alpha$-WFAT introduces the relaxation into federated adversarial training, which is orthogonal to and compatible with the previous optimization methods.

**Adversarial Training**   As one of the defensive methods (Papernot et al., 2016; Gao et al., 2021), Adversarial Training (Madry et al., 2018; Zhang et al., 2019; Jiang et al., 2020; Chen et al., 2021) is to improve the robustness of machine learning models. The classical AT (Madry et al., 2018) is built upon on a min-max formula to optimize the worst case, *e.g.,* the adversarial example near the natural example (Goodfellow et al., 2015). Zhang et al. (2019) decomposed the prediction error for adversarial examples as the sum of the natural error and the boundary error, and proposed TRADES to balance the classification performance between the natural and adversarial examples. Wang et al. (2020c) further explored the influence of the misclassified examples on the robustness, and proposed MART that emphasizes the minimization of the misclassified examples to boost the AT. Zhang et al. (2020) investigated the "benign adversarial examples" in AT and further improve the natural performance of robust model. In this work, our $\alpha$-WFAT framework leverages the client-level measure to alleviate the heterogeneous issue in the straightforward combination of adversarial training and federated learning. It is compatible to further incorporate those centralized adversarial training methods to improve the model performance.

**Federated Adversarial Training.**   Recently, several works have made the exploration on the Adversarial Training in the context of Federated Learning, which consider the data privacy and the robustness in one framework. To our best knowledge, Zizzo et al. (2020) take the first trial to study the feasibility of extending Federated Learning (McMahan et al., 2017) with the standard AT on both IID and Non-IID settings. Empirically, they found that there was a large performance gap existing between the distributed and the centralized adversarial training, especially on the Non-IID data. Shah et al. (2021) designed a dynamic schedule for the local training to pursue a larger robustness under the constrained communication budget of Federated Learning. Hong et al. (2021) explored how to effectively propagate the adversarial robustness when only limited clients in Federated Learning have the sufficient computational budget to afford AT. Although previous works have investigated to solve the challenges about the constrained communication or computational budget, one critical issue that affects the performance when combined Adversarial Training with Federated Learning has received only few discussion. In this work, we consider such a basic issue (see Figure 1) in the straightforward combination of AT with Federated Learning and introduce our solution to this problem.

## 3   PRELIMINARY

In this section, we will briefly formalize the notations of Adversarial Training (Goodfellow et al., 2015; Madry et al., 2018) and Federated Learning as well as FedAvg (McMahan et al., 2017).

### 3.1   ADVERSARIAL TRAINING

Let $(\mathcal{X}, d_\infty)$ denote the input feature space $\mathcal{X}$ with the infinity distance metric $d_\infty(x, \tilde{x}) = \|x - \tilde{x}\|_\infty$, and $\mathcal{B}_\epsilon[x] = \{\tilde{x} \in \mathcal{X} \mid d_\infty(x, \tilde{x}) \leq \epsilon\}$ be the closed ball of radius $\epsilon > 0$ centered at $x$ in $\mathcal{X}$. Dataset $S = \{(x_n, y_n)\}_{n=1}^N$, where $x_n \in \mathcal{X}$ and $y_n \in \mathcal{Y} = \{0, 1, ..., C-1\}$. The objective function of the standard adversarial training (AT) (Madry et al., 2018) is defined as follows,

$$\min_{f_\theta \in \mathcal{F}} \frac{1}{N} \sum_{n=1}^N \max_{\tilde{x}_n \in \mathcal{B}_\epsilon[x_n]} \ell(f_\theta(\tilde{x}_n), y_n), \tag{1}$$

where $\tilde{x}$ is the *most adversarial data* within the $\epsilon$-ball centered at $x$, $f_\theta(\cdot) : \mathcal{X} \to \mathbb{R}^C$ is a score function, $\ell : \mathbb{R}^C \times \mathcal{Y} \to \mathbb{R}$ is a composition of a base loss $\ell_B : \Delta^{C-1} \times \mathcal{Y} \to \mathbb{R}$ (*e.g.,* the Cross-Entropy loss) and an inverse link function $\ell_L : \mathbb{R}^C \to \Delta^{C-1}$ (*e.g.,* the Softmax). Here, $\Delta^{C-1}$ is the corresponding probability simplex that yields $\ell(f_\theta(\cdot), y) = \ell_B(\ell_L(f_\theta(\cdot)), y)$. For the inner-maximization of Eq. (1), the multi-step projected gradient descent (PGD) (Madry et al., 2018) is usually applied to find the most adversarial samples, which are then used for the outer-minimization.

Figure 2: Left panel: locally learned decision boundary on Client A; Middle left panel: locally learned decision boundary on Client B; Middle right panel: globally aggregated decision boundary based on *FedAvg*; Right panel: globally aggregated decision boundary based on $\alpha$-*WFAT*. Note that, the distance between the correctly classified adversarial examples and the decision boundary (*i.e.,* the bold line) can approximately reflect the client loss, and shows that $\ell_{client\ A} > \ell_{client\ B}$. Then, selectively treating two client models in the aggregation can acquire a better global model (e.g., the fourth panel), which is consistent with the intuition of Eq. (3).

## 3.2 FEDERATED LEARNING

Let $\mathcal{D}_k$ denotes a finite set of samples from the $k$-th client, and in each round, a set of datasets $\{\mathcal{D}_k\}_{k=1}^K$ from $K$ clients are involved into the training. The objective of Federated Learning is to learn a machine learning model without any exchange of the training data between the clients and the server. The current popular strategy, namely FedAvg, is introduced by McMahan et al. (2017), where the clients collaboratively send the locally trained model weights $\theta_k$ to the server for the global average aggregation. Concretely, each client runs on a local copy of the global model (parameterized by $\theta^t$ in the $t$-th round) with its local data to optimize the objective like Eq. (1). Then, the server receives their updated model weights $\{\theta_k^t\}_{k=1}^K$ of all clients and performs the following aggregation

$$\theta^{t+1} = \frac{1}{N} \sum_{k=1}^K N_k \theta_k^t, \tag{2}$$

where $N_k$ denotes the number of the samples in $\mathcal{D}_k$ and $N = \sum_{k=1}^K N_k$. Then, the weights $\theta^{t+1}$ for the global model will be sent back to each client for another lifecycle. After the sufficient rounds of such a periodic synchronization and aggregation, we expect the stationary point of Federated Learning will approximately approach to or have a small gap with that from the centralized counterpart.

## 4 $\alpha$-WEIGHTED FEDERATED ADVERSARIAL TRAINING

In the following sections, we will first introduce our motivation for Federated Adversarial Training through analyzing the challenges brought by the straightforward combination. Then, we relax the inner-maximization problem of Adversarial Training as a lower bound by an $\alpha$-weighted decomposition, and present the theoretical understanding of such a relaxation. Finally, we will present the $\alpha$-Weighted Federated Adversarial Training and the corresponding analysis.

## 4.1 MOTIVATION

For the general FAT (Zizzo et al., 2020), in each round, the clients will receive the latest model from the server. Then, they conduct Adversarial Training with the local data and send their optimized model parameters to the server. The server will aggregate the parameters of all clients into a global model with FedAvg or other methods in Federated Learning. However, considering the characteristics of Federated Learning and Adversarial Training, it naturally brings the following critical challenge.

*The inner-maximization can enlarge the data heterogeneity in Federated Learning.* In terms of the combination of adversarial training and federated learning, the key point is that such adversarial examples generated by the inner-maximization exacerbate the heterogeneity, which induces the performance deterioration under the model aggregation methods. As the Figure 2 shows, the client A with large adversarial loss might make a dominating effect on the convergence of the global model, given the current model aggregation methods treat all client models indiscriminately. Thus, it might be better to selectively weight the client models in the model aggregation.

### 4.2 $\alpha$-WEIGHTED RELAXATION OF DECOMPOSED ADVERSARIAL TRAINING

As previous analysis, the inner-maximization of Adversarial Training is not very compatible with Federated Learning due to the exacerbation of heterogeneity when combing them together. An idea to alleviate this problem is building a mediating function that keeps the original goal but is friendly to two paradigms. In this section, we present one possible solution to this intuition, which tries to find a relaxation of the inner-maximization in Adversarial Training. Formally, we decompose the inner-maximization objective into the independent $K$ populations that corresponds to the $K$ clients in Federated Learning, and relaxes it into a lower bound by the $\alpha$-weighted mechanism as follows,

$$
\begin{aligned}
\mathcal{L}_{AT} &= \frac{1}{N} \sum_{n=1}^{N} \max_{\tilde{x}_n \in \mathcal{B}_\epsilon[x_n]} \ell(f(\tilde{x}_n), y_n) = \sum_{k=1}^{K} \frac{N_k}{N} \underbrace{\left( \frac{1}{N_k} \sum_{n=1}^{N_k} \max_{\tilde{x}_n \in \mathcal{B}_\epsilon[x_n]} \ell(f(\tilde{x}_n^k), y_n^k) \right)}_{\mathcal{L}_k} \\
&\geq (1+\alpha) \sum_{k=1}^{\widehat{K}} \frac{N_{\phi(k)}}{N} \mathcal{L}_{\phi(k)} + (1-\alpha) \sum_{k=\widehat{K}+1}^{K} \frac{N_{\phi(k)}}{N} \mathcal{L}_{\phi(k)} \quad \text{s.t. } \alpha \in [0,\ 1),\ \widehat{K} \leq \frac{K}{2} \\
&\doteq \mathcal{L}^\alpha(\widehat{K}),
\end{aligned}
\tag{3}
$$

where $\phi(\cdot)$ is a function which maps the index to the original population sorted by $\{\frac{N_k}{N} \mathcal{L}_k\}$ in an ascending order. The following theorems provide us more analysis about the $\alpha$-weighted relaxation.

**Theorem 4.1.** *$\mathcal{L}^\alpha(\widehat{K})$ is monotonically decreasing w.r.t. both $\alpha$ and $\widehat{K}$, i.e., $\mathcal{L}^{\alpha_1}(\widehat{K}) < \mathcal{L}^{\alpha_2}(\widehat{K})$ if $\alpha_1 > \alpha_2$ and $\mathcal{L}^\alpha(\widehat{K}_1) < \mathcal{L}^\alpha(\widehat{K}_2)$ if $\widehat{K}_1 > \widehat{K}_2$. Specifically, $\mathcal{L}^\alpha(\widehat{K})$ recovers $\mathcal{L}$ of adversarial training when $\alpha$ achieves 0, and $\mathcal{L}^\alpha(\widehat{K})$ relaxes $\mathcal{L}$ to a lower bound objective by increasing $\widehat{K}$ and $\alpha$.*

We can flexibly emphasize the importance of partial populations by setting the proper hyperparameters, alleviating the evenly averaging of harsh heterogeneous updates in FedAvg (McMahan et al., 2017).

**Theorem 4.2.** *Assume the loss function $\ell(\cdot, \cdot)$ in Eq. (3) satisfies the Lipschitzian smoothness condition w.r.t. the model parameter $\theta$ and the training sample $x$, and is $\lambda$-strongly concave for all $x$, and $\mathbb{E}\left[ ||\nabla \mathcal{L}^\alpha(\widehat{K}) - \nabla_\theta \ell(f(\tilde{x}), y)||_2^2 \right] \leq \delta^2$, where $\tilde{x}$ is the adversarial example. Then, after the sufficient $T$-step optimization i.e., $T \geq \frac{L\Delta}{\delta^2}$, for $\alpha$-weighted relaxation of decomposed Adversarial Training with the constant stepsize $\sqrt{\frac{\Delta}{LT\sigma^2}}$ in PGD, we have the following convergence property,*

$$
\frac{1}{T} \sum_t^T \mathbb{E}\left[ \left|\left| \nabla \mathcal{L}^\alpha(\widehat{K})|_{\theta^t} \right|\right|_2^2 \right] \leq \left( 1 + \alpha \frac{\frac{1}{T}\sum_t^T \xi^{(t)}}{N} \right) \left( \frac{4L_{\theta x}^2 \epsilon}{\lambda} + 4\delta \sqrt{\frac{L\Delta}{T}} \right),
\tag{4}
$$

*where $L = L_{\theta\theta} + \frac{L_{\theta x} L_{x\theta}}{\lambda}$ defined by the Lipschitzian constraints, $\Delta \geq \mathcal{L}^\alpha(\widehat{K})|_{\theta^0} - \inf_\theta \mathcal{L}^\alpha(\widehat{K})$ and $\xi^{(t)} = \sum_k^{\widehat{K}} N_{\phi(k)}^{(t)} - \sum_{\widehat{K}+1}^{K} N_{\phi(k)}^{(t)}$ meaning the accumulative counting difference of the $t$-th step.*

When $\alpha = 0$, Eq.(3) recovers the original loss of Adversarial Training, and the first part in the RHS of Eq. (4) goes to 1 that recovers the convergence rate of Adversarial Training (Sinha et al., 2018). When $\alpha \to 1$, Eq.(3) becomes more biased, while simultaneously the straightforward benefit is that we can achieve a faster convergence in Eq. (4) if $\frac{1}{T}\sum_t^T \xi^{(t)} < 0$, i.e., $\left( 1 + \alpha \frac{\frac{1}{T}\sum_t^T \xi^{(t)}}{N} \right) < 1$. Actually, this is possible when the sample number is approximately similar among all clients and the top-1 choice easily has $-N < \xi^{(t)} = N_{\phi(1)}^{(t)} - \sum_{k=2}^{K} N_{\phi(k)}^{(t)} < 0$ in each optimization step. Especially, a larger $\alpha$ has a faster convergence in this case. Therefore, the $\alpha$-weighted relaxation of decomposed Adversarial Training provides us a trade-off between maintaining the robustness from the standard robust training and achieving the faster convergence with some biased approximation by $\alpha$ and $\widehat{K}$.

### 4.3 $\alpha$-WEIGHTED FEDERATED ADVERSARIAL TRAINING

Inspired by the previous analysis of $\alpha$-weighted relaxation, we propose an $\alpha$-*Weighted Federated Adversarial Training* to combine Adversarial Training and Federated Learning. The intuition is

---

**Algorithm 1** $\alpha$-Weighted Federated Adversarial Training ($\alpha$-WFAT)

---

**Input:** number of clients: $K$, number of communication rounds: $T$, number of client training epochs
per round: $E$, initial server's model parameter: $\theta^0$, hyper-parameter for aggregation: $\alpha$, number
of enhanced clients: $\widehat{K}$;
**Output:** a globally robust model with parameter $\theta^T$;
 1: **for** t = 0, ..., $T - 1$ **do**
 2:      **Clients:** [ perform adversarial training]
 3:      **for** client $k = 1, \ldots, K$ **do**
 4:          $\theta_k^t, \mathcal{L}_k = \text{AT}(\theta_k^t, E)$ (Madry et al., 2018)
 5:      **end for**
 6:      **Server:** [ performs aggregation over weight updates]
 7:      $\mathcal{L}_{all} \leftarrow [\frac{N_1}{N}\mathcal{L}_1, \frac{N_2}{N}\mathcal{L}_2, \ldots, \frac{N_K}{N}\mathcal{L}_K]$
 8:      $\mathcal{L}_{sorted} \leftarrow \text{Ascending\_Sort}(\mathcal{L}_{all})$
 9:      $\forall k, \ P_k \leftarrow (1+\alpha) \cdot \mathbb{1}(\frac{N_k}{N}\mathcal{L}_k \le \mathcal{L}_{sorted}[\widehat{K}]) + (1-\alpha) \cdot \mathbb{1}(\frac{N_k}{N}\mathcal{L}_k > \mathcal{L}_{sorted}[\widehat{K}])$;
10:      $\theta^{t+1} = \frac{1}{\sum_{k=1}^K P_k N_k} \sum_{k=1}^K P_k N_k \theta_k^t$;          $\triangleright$ $\alpha$-weighted mechanism
11: **end for**

---

applying the $\alpha$-weighted mechanism into the inner-maximization in FAT, formalized as follows,

$$\min \mathcal{L}_{\alpha-WFAT} = \min_{f_\theta \in \mathcal{F}} \frac{1}{\sum_k^K P_k N_k} \sum_{k=1}^K P_k N_k \underbrace{\left( \frac{1}{N_k} \sum_{n=1}^{N_k} \max_{\tilde{x}_n \in \mathcal{B}_\epsilon[x_n]} \ell(f_\theta(\tilde{x}_n^k), y_n^k) \right)}_{\mathcal{L}_k}, \quad (5)$$

where $P_k = (1+\alpha) \cdot \mathbb{1}(\frac{N_k}{N}\mathcal{L}_k \le \mathcal{L}_{sorted}[\widehat{K}]) + (1-\alpha) \cdot \mathbb{1}(\frac{N_k}{N}\mathcal{L}_k > \mathcal{L}_{sorted}[\widehat{K}])$ denotes the weight
assigned to the $k$-th client based on the ascending sort of weighted client losses compared with the
$\widehat{K}$-th one. We summarize the procedure of $\alpha$-WFAT in Algorithm 1, which consists of multi-round
iterations between the local training on the client side and the global aggregation on the server side.

Concretely, on the client side, after downloading the global model parameter from the server, each
client will perform the Adversarial Training on its local data. At the same time, the client loss on the
adversarial examples is also recorded, which acts as the soft-indicator of the local bias induced by the
radical adversarial generation. Then, when the training steps reach to the condition, the client will
upload its model parameter and the loss to the server. On the server side, after collecting the model
parameters $\{\theta_k\}_{k=1}^K$ and the losses $\{\mathcal{L}_k\}_{k=1}^K$ of all clients, it will first sort the losses in an ascending
order to find the top-$\widehat{K}$ clients. Based on that, the global model parameters will be aggregated by
the $\alpha$-weighted mechanism in which the model parameters of the top-$\widehat{K}$ clients are upweighted with
$(1 + \alpha)$ and the remaining is downweighted by $(1 - \alpha)$. For some atypical layers (Li et al., 2021b)
*e.g.,* BN, it is outside the scope of this paper and we keep the aggregation same as FedAvg. Note that,
one interesting point in $\alpha$-WFAT is the top-$\widehat{K}$ clients with the higher weights are not fixed, and they
can be routing among all clients. In Figure 3, we trace this dynamic of $\alpha$-WFAT in one experiment.

In the following, we provide the theoretical analysis of $\alpha$-WFAT on the convergence in the context of
Federated Learning (Li et al., 2019), which is slightly different from previous centralized counterpart.

**Theorem 4.3.** *Assume the loss function $\ell(\cdot, \cdot)$ in Eq. (5) is L-smooth and $\lambda$-strongly concave w.r.t.
the model parameter $\theta$, and the expected norm and the variance of the stochastic gradient in each
client respectively satisfy $\mathbb{E}\left[||\nabla_\theta \ell(f(\tilde{x}^k), y^k)||_2^2\right] \le \varsigma^2$ and $\mathbb{E}\left[||\nabla_\theta \ell(f(\tilde{x}^k), y^k) - \nabla_\theta \mathcal{L}_k||_2^2\right] \le \delta_k^2$.
Let $\kappa = \frac{L}{\lambda}, \gamma = \max\{8\kappa, E\}$ where $E$ is the iteration number of the local Adversarial Training with
the learning rate $\eta_t = \frac{2}{\lambda(\gamma+t)}$. Then, after the sufficient $T$-step communication rounds for $\alpha$-WFAT,
we have the following asymptotics to the optimal point,*

$$\mathbb{E}[\mathcal{L}_{\alpha-WFAT}] - \mathcal{L}^* \le \frac{\kappa}{\gamma + T - 1}\left( \frac{2B}{\lambda} + \frac{\lambda\gamma}{2}\mathbb{E}\left[||\theta^0 - \theta^*||^2\right] \right), \quad (6)$$

*where $\mathcal{L}^*$ is the minimum value of $\mathcal{L}_{\alpha-WFAT}$, $\theta^*$ is the optimal model parameter, and*

$$B = \sum_k^K \left( \frac{P_k^{(T)}}{1 + \alpha\frac{\xi^{(T)}}{N}} \right)^2 \left( \frac{N_k}{N}\delta_k \right)^2 + 6L\left( \mathcal{L}^* - \sum_k^K \frac{P_k^{(T)}}{1 + \alpha\frac{\xi^{(T)}}{N}} \frac{N_k}{N}\mathcal{L}_k^* \right) + 8(E - 1)^2\varsigma^2.$$

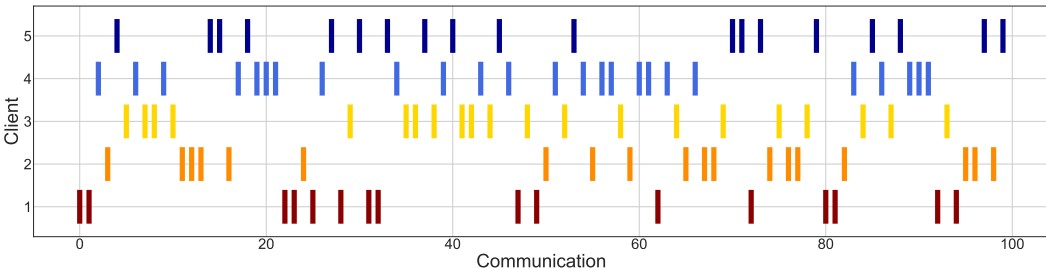

Figure 3: The index of the top-$\widehat{K}$ clients with the small losses in $\alpha$-WFAT ($\alpha = 1/6$, $\widehat{K} = 1$) in each communication round on CIFAR-10. We can see that it is dynamically routing among all clients.

When $\alpha = 0$, we have $P_k^{(t)} = 1$ and Eq.( 6) becomes the convergence rate of FedAvg on non-IID data (Li et al., 2019). Different from Theorem 4.2 that concludes in the centralized training setting, when $\alpha \to 1$, the convergence is indefinite compared to the standard FAT, since the emerging terms in $B$, *i.e.*, $\left( \frac{P_k^{(T)}}{1 + \alpha \frac{\xi^{(T)}}{N}} \right)^2$ and $\frac{P_k^{(T)}}{1 + \alpha \frac{\xi^{(T)}}{N}}$, are acted as the scalar timing by the personalized variance bound $\delta_k^2$ and the local optimum $\mathcal{L}_k^*$ of each client. One possible case is when the optimization approaches to the optimal parameter $\theta^*$, $\delta_k^2$ can be in a smaller scale relative to the scale of $\mathcal{L}_k^*$. In this case, the increment of the first term of $B$ can be totally counteracted by the loss of second term of $B$ so that in sum $B$ becomes smaller. Then, we can have a tighter upper bound for $\alpha$-WFAT in Eq.( 6) to achieve a faster convergence than FAT. The completed proof of Theorem 4.3 can refer to Appendix C and the experiments in the following section will confirm $\alpha$-WFAT can reach to a more robust optimum.

## 5 EXPERIMENTS

In this section, we will provide an in-depth analysis of $\alpha$-WFAT and empirically verify its efficiency compared with the current methods on a range of IID and non-IID datasets.

### 5.1 EXPERIMENTAL SETUP

**Dataset.** We conduct the experiments on three benchmark datasets, *i.e., CIFAR-10*, *CIFAR-100* (Krizhevsky, 2009) and *SVHN* (Netzer et al., 2011) for Federated Adversarial Training. For the IID scenario, we just randomly and evenly distribute the samples to each client. For the Non-IID scenario, we follow McMahan et al. (2017); Shah et al. (2021) to partition the training data based on their labels. To be specific, a skew parameter $s$ is utilized in the data partition introduced by Shah et al. (2021), which enables $K$ clients to get a majority of the data samples from a subset of classes. We denote the set of all classes in a dataset as $\mathcal{Y}$ and create $\mathcal{Y}_k$ by dividing all the class labels equally among $K$ clients. Accordingly, we split the data across $K$ clients that each client has $(100 - (K - 1) \times s)\%$ of data for the class in $\mathcal{Y}_k$ and $s\%$ of data in other split sets. The detailed training and evaluation settings can refer to Appendix E.

### 5.2 ABLATION STUDY

In this part, we conduct various experiments on *CIFAR-10* to visualize the characteristics of $\alpha$-WFAT.

**Non-AT vs. AT.** In the left two panels of Figure 4, we respectively apply our $\alpha$-weighted mechanism to Federated Standard Training ($\alpha$-WFST) and Federated Adversarial Training ($\alpha$-WFAT) under $\widehat{K} = 1$. We also consider both FedAvg and FedProx in this experiment to guarantee the universality. From the curves, we can see that $\alpha$-WFST has the negative effect on the natural accuracy, while $\alpha$-WFAT consistently improve the robust accuracy both based on FedAvg or FedProx. This indicates our $\alpha$-weighted mechanism is tailored for the inner-maximization of Federated Adversarial Training instead of the outer-minimization considered by other federated optimization methods.

**Impact of $\alpha$ and $\widehat{K}$.** To study the effect of hyperparameter in $\alpha$-WFAT, we conduct several ablation experiments to verify the model performance. Regarding the experiments of $\alpha$, we set the client

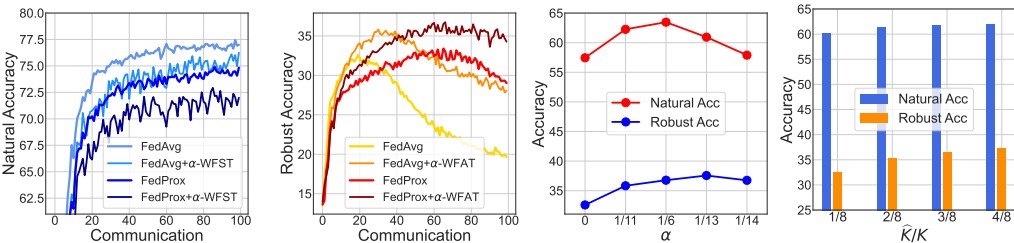

Figure 4: Ablation study on $\alpha$-WFAT. Left two panels: comparison between Federated Standard Training and Federated Adversarial Training respectively in combination with the $\alpha$-weighted mechanism, *i.e.,* ($\alpha$-WFST) vs. ($\alpha$-WFAT). Right two panels: the natural accuracy and the robust accuracy of $\alpha$-WFAT with different $\alpha$ and different $\widehat{K}$ on CIFAR-10.

number $K = 5$ and $\widehat{K} = 1$ to upweight/downweight the client models in each communication rounds. The right middle panel of Figure 4 shows $\alpha \in (0, 1/6]$ can significantly improve the robust accuracy and the natural accuracy, while a larger $\alpha$ might be inappropriate to the natural accuracy. Regarding the choice of $\widehat{K}$, we specially set $K = 8$ in this experiment to span the range of $\widehat{K}$ due to the constraint $\widehat{K} <= K/2$. The right panel of Figure 4 presents the accuracy of $\alpha$-WFAT with increasing $\widehat{K}$.

According to the plot, both the natural accuracy and the robust accuracy are improved even with larger $\widehat{K}$, which shows the effect of $\widehat{K}$ on the relaxation of inner-maximization in the proposed $\alpha$-WFAT. In addition, we conduct the experiments about emphasizing/de-emphasizing the client with smallest adversarial loss by adjusting the $\alpha$, the results (in Appendix E.1) confirmed that the relaxation introduced by our $\alpha$-WFAT is needed to improve performance, and is consistent with our previous analysis.

Table 1: Test accuracy (%) on *CIFAR-10* (Non-IID) with different Adversarial Training methods on the local client.

| Methods | | Natural | PGD-20 | CW$_\infty$ |
|---------|------|---------|--------|-------------|
| AT | FAT | 57.45% | 32.58% | 30.52% |
| | $\alpha$-**WFAT** | **62.34%** | **35.59%** | **33.06%** |
| TRADES | FAT | 64.00% | 31.64% | 28.95% |
| | $\alpha$-**WFAT** | **65.26%** | **35.10%** | **31.80%** |
| MART | FAT | 56.29% | 36.27% | 32.41% |
| | $\alpha$-**WFAT** | **58.41%** | **38.90%** | **34.67%** |

**Different AT methods.** In Table 1, we validate the combination of $\alpha$-weighted mechanism and different Adversarial Training methods (*i.e.,* TRADES (Zhang et al., 2019) and MART (Wang et al., 2020c)), where we switch different local Adversarial Training methods on the client side. Through the comparison with FAT, the results show that $\alpha$-WFAT can consistently improve both the natural performance and the robust performance, and is general to the state-of-the-art Adversarial Training works under Federated Learning scenarios.

## 5.3 PERFORMANCE EVALUATION

In this section, we compare our $\alpha$-WFAT with FAT on various benchmark datasets to demonstrate its effectiveness. Specifically, we validate both the non-IID and IID settings with three representative Federated optimization methods *i.e.,* FedAvg, FedProx and Scalffold. Besides, a centralized AT baseline is provided as the reference of the IID setting. Note that, there is no such a baseline for the non-IID setting, since the centralized case is one distribution which is incomparable and meaningless. Considering the sensitivity of data selection in Non-IID settings, we also report the results with Mean and Std values in Appendix E.2 after running experiments for multiple times.

According to Table 2 on CIFAR-10, we can find that $\alpha$-WFAT significantly outperforms FAT on the Non-IID data in terms of both the natural accuracy ($\sim$2%-6%) and the robust accuracy ($\sim$2%-5%). For the IID data, our method acquires a similar improvement on the robust accuracy without the deterioration of the natural accuracy. The reason might be because even the data is IID, Adversarial Training can still drive the independently-initialized overparameterized network (Allen-Zhu & Li, 2020) on each client side towards at the robust overfitting of different directions, yielding the model heterogeneity. Thus, the proper relaxation to the inner-maximization makes Adversarial Training more compatible with Federated Learning. Another interesting observation is that Federated Adversarial Training shows better performance than centralized Adversarial Training in the case of the IID setting. This gain could be from the distributed training paradigm that helps Adversarial Training converge

Table 2: Performance on three benchmark datasets under different federated optimization methods.

| Setting | | Non-IID | | | | | IID | | | | |
|---|---|---|---|---|---|---|---|---|---|---|---|
| CIFAR-10 | | Natural | FGSM | PGD-20 | $CW_\infty$ | AA | Natural | FGSM | PGD-20 | $CW_\infty$ | AA |
| Centralized AT | | - | - | - | - | - | 66.47% | 47.68% | 38.18% | 37.04% | 34.48% |
| FedAvg | FAT | 57.45% | 39.44% | 32.58% | 30.52% | 29.20% | **69.35%** | 48.45% | 37.43% | 35.72% | 33.96% |
| | $\alpha$-WFAT | **63.44%** | **45.13%** | **37.17%** | **33.99%** | **32.36%** | 67.43% | **50.33%** | **42.78%** | **37.91%** | **36.20%** |
| FedProx | FAT | 60.44% | 41.59% | 33.84% | 31.29% | 30.02% | 66.91% | 46.70% | 37.14% | 34.54% | 32.68% |
| | $\alpha$-WFAT | **62.51%** | **44.29%** | **36.75%** | **33.82%** | **31,98%** | **68.31%** | **48.40%** | **42.41%** | **37.25%** | **35.97%** |
| Scaffold | FAT | 62.81% | 43.61% | 34.13% | 32.53% | 30.95% | 68.27% | 49.25% | 39.33% | 37.31% | 35.30% |
| | $\alpha$-WFAT | **64.12%** | **46.05%** | **37.35%** | **34.78%** | **33.32%** | **71.36%** | **50.42%** | **43.83%** | **39.12%** | **35.47%** |
| CIFAR-100 | | Natural | FGSM | PGD-20 | $CW_\infty$ | AA | Natural | FGSM | PGD-20 | $CW_\infty$ | AA |
| Centralized AT | | - | - | - | - | - | 35.81% | 23.09% | 18.64% | 16.48% | 15.42% |
| FedAvg | FAT | 31.07% | 19.60% | 16.16% | 13.37% | 12.47% | 38.35% | 23.37% | 18.44% | 16.63% | 15.45% |
| | $\alpha$-WFAT | **35.17%** | **21.26%** | **16.72%** | **13.91%** | **12.83%** | **38.43%** | **23.76%** | **18.82%** | **16.71%** | **15.62%** |
| FedProx | FAT | 33.33% | 20.20% | 16.08% | 13.76% | 12.72% | 37.18% | 22.29% | 18.16% | 16.33% | 15.29% |
| | $\alpha$-WFAT | **34.30%** | **20.82%** | **16.74%** | **13.84%** | **12.88%** | **37.37%** | **23.11%** | **18.43%** | **16.36%** | **15.45%** |
| Scaffold | FAT | **41.17%** | 25.17% | 20.01% | 16.74% | 15.49% | **42.42%** | 26.79% | 21.18% | 18.89% | 17.63% |
| | $\alpha$-WFAT | 41.07% | **25.55%** | **20.40%** | **16.79%** | **15.59%** | 42.08% | **27.18%** | **22.26%** | **19.34%** | **18.03%** |
| SVHN | | Natural | FGSM | PGD-20 | $CW_\infty$ | AA | Natural | FGSM | PGD-20 | $CW_\infty$ | AA |
| Centralized AT | | - | - | - | - | - | 92.39% | 89.75% | 72.73% | 72.31% | 70.93% |
| FedAvg | FAT | 91.24% | 87.95% | 68.87% | 67.89% | 66.54% | **93.52%** | **90.68%** | 72.24% | 71.22% | 70.08% |
| | $\alpha$-WFAT | **91.25%** | **88.28%** | **71.72%** | **69.79%** | **68.62%** | 92.75% | 90.06% | **74.37%** | **72.34%** | **71.27%** |
| FedProx | FAT | 90.92% | 87.50% | 68.44% | 67.18% | 65.94% | 93.54% | 90.66% | 72.53% | 71.42% | 70.21% |
| | $\alpha$-WFAT | **91.25%** | **88.15%** | **71.54%** | **69.53%** | **68.47%** | **93.59%** | **90.80%** | **74.66%** | **72.67%** | **71.48%** |
| Scaffold | FAT | 89.95% | 87.23% | 68.66% | 67.23% | 66.65% | 93.80% | 91.00% | 73.26% | 72.05% | 70.80% |
| | $\alpha$-WFAT | **90.20%** | **87.81%** | **71.39%** | **68.81%** | **67.88%** | **93.92%** | **91.28%** | **75.96%** | **74.05%** | **72.88%** |

to the more robust optimum by the divide-and-conquer mechanism. This might enlighten the more explore in Adversarial Training to improve the model robustness via Federated Learning.

On *CIFAR-100* and *SVHN*, we can find the similar improvement in Table 2 as that of *CIFAR-10* under three types of federated optimization methods. Nevertheless, $\alpha$-WFAT only becomes superior in terms of the robust accuracy but comparable with FAT in terms of the natural accuracy especially on CIFAR-100. It indicates that the inner-maximization of Adversarial Training when combined with Federated Learning mainly affects the model robustness, and thus the $\alpha$-weighted relaxation correspondingly helps the model converge to a more robust optimum.

## 6  CONCLUSION

In this work, we explore the performance deterioration in the straightforward combination of Adversarial Training with Federated Learning. To alleviate the potential radical optimization, we apply an $\alpha$-weighted relaxation into Adversarial Training to relax the inner-maximization. Based on this $\alpha$-weighted mechanism, we further propose $\alpha$-Weighted Federated Adversarial Training ($\alpha$-WFAT). We provide the theoretical analysis and empirical evidences to understand the proposed simple but effective method. The experimental results under different settings confirm the effectiveness of $\alpha$-WFAT. Nevertheless, we only move a small step on the heterogeneous issue in the combination of two paradigms and more issues in their cross field could be further explored in the future.

## 7 ETHICS STATEMENT

This paper does not raise any ethics concerns. This study does not involve any human subjects, practices to data set releases, potentially harmful insights, methodologies and applications, potential conflicts of interest and sponsorship, discrimination/bias/fairness concerns, privacy and security issues, legal compliance, and research integrity issues.

## 8 REPRODUCIBILITY STATEMENT

To ensure the reproducibility of experimental results, we will provide a link for an anonymous repository about the source codes of this paper in the discussion forums.

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

APPENDIX

## A PROOF OF EQ. (3) AND THEOREM 4.1

We proof the Eq. (3) and Theorem 4.1 in this section.

Recall the $\alpha$-weighted relaxation for the inner-maximization objective decomposition with $K$ independent populations as follows,

$$
\mathcal{L}_{AT} = \frac{1}{N} \sum_{n=1}^{N} \max_{\tilde{x}_n \in \mathcal{B}_\epsilon[x_n]} \ell(f(\tilde{x}_n), y_n) = \sum_{k=1}^{K} \frac{N_k}{N} \underbrace{\left( \frac{1}{N_k} \sum_{n=1}^{N_k} \max_{\tilde{x}_n \in \mathcal{B}_\epsilon[x_n]} \ell(f(\tilde{x}_n^k), y_n^k) \right)}_{\mathcal{L}_k}
$$

$$
\geq (1+\alpha) \sum_{k=1}^{\widehat{K}} \frac{N_{\phi(k)}}{N} \mathcal{L}_{\phi(k)} + (1-\alpha) \sum_{k=\widehat{K}+1}^{K} \frac{N_{\phi(k)}}{N} \mathcal{L}_{\phi(k)} \quad \text{s.t. } \alpha \in [0,\ 1),\ \widehat{K} \leq \frac{K}{2} \tag{7}
$$

$$
\doteq \mathcal{L}^\alpha(\widehat{K}),
$$

where $\phi(\cdot)$ is a function which maps the index to the original population group sorted by $\{\frac{N_k}{N}\mathcal{L}_k\}$ in an ascending order.

*proof of Eq. (3).* The deduction of the inequality in Eq. (7) can be formulated in the following. Given $\alpha \in [0,1)$ and $\widehat{K} \leq \frac{K}{2}$ with the population sorted by $\{\frac{N_k}{N}\mathcal{L}_k\}$ in an ascending order, we have $\sum_{k=1}^{\widehat{K}} \frac{N_{\phi(k)}}{N}\mathcal{L}_{\phi(k)} \leq \sum_{k=\widehat{K}+1}^{K} \frac{N_{\phi(k)}}{N}\mathcal{L}_{\phi(k)}$. Then, we have the following relationship by subtraction,

$$
\sum_{k=1}^{\widehat{K}} \frac{N_{\phi(k)}}{N}\mathcal{L}_{\phi(k)} + \sum_{k=\widehat{K}+1}^{K} \frac{N_{\phi(k)}}{N}\mathcal{L}_{\phi(k)} - (1+\alpha) \sum_{k=1}^{\widehat{K}} \frac{N_{\phi(k)}}{N}\mathcal{L}_{\phi(k)} - (1-\alpha) \sum_{k=\widehat{K}+1}^{K} \frac{N_{\phi(k)}}{N}\mathcal{L}_{\phi(k)}
$$

$$
= \alpha \cdot \left( \sum_{k=\widehat{K}+1}^{K} \frac{N_{\phi(k)}}{N}\mathcal{L}_{\phi(k)} - \sum_{k=1}^{\widehat{K}} \frac{N_{\phi(k)}}{N}\mathcal{L}_{\phi(k)} \right) \geq 0. \tag{8}
$$

$\square$

*proof of Theorem 4.1.* It can be naturally proved by Eq. (8). If $\alpha_1 > \alpha_2$, then we have,

$$
\mathcal{L}^{\alpha_1}(\widehat{K}) - \mathcal{L}^{\alpha_2}(\widehat{K}) = (1+\alpha_1) \sum_{k=1}^{\widehat{K}} \frac{N_{\phi(k)}}{N}\mathcal{L}_{\phi(k)} + (1-\alpha_1) \sum_{k=\widehat{K}+1}^{K} \frac{N_{\phi(k)}}{N}\mathcal{L}_{\phi(k)}
$$

$$
- (1+\alpha_2) \sum_{k=1}^{\widehat{K}} \frac{N_{\phi(k)}}{N}\mathcal{L}_{\phi(k)} - (1-\alpha_2) \sum_{k=\widehat{K}+1}^{K} \frac{N_{\phi(k)}}{N}\mathcal{L}_{\phi(k)} \tag{9}
$$

$$
= (\alpha_1 - \alpha_2) \left( \sum_{k=1}^{\widehat{K}} \frac{N_{\phi(k)}}{N}\mathcal{L}_{\phi(k)} - \sum_{k=\widehat{K}+1}^{K} \frac{N_{\phi(k)}}{N}\mathcal{L}_{\phi(k)} \right) \leq 0
$$

Similarly, we can prove $\mathcal{L}^\alpha(\widehat{K}_1) < \mathcal{L}^\alpha(\widehat{K}_2)$ if $\widehat{K}_1 > \widehat{K}_2$. $\square$

## B PROOF OF THEOREM 4.2

Based on the convergence of Adversarial Training (Sinha et al., 2018), we proof Theorem 4.2 in this section.

*proof of Theorem 4.2.* Let $c : \mathcal{X} \times \mathcal{X} \to \mathbb{R}_+ \cup \{\infty\}$, where $c(x, x_0)$ is the "cost" for an adversary to perturb $x_0$ to $x$. Let $f(\theta, x; x_0) = \ell(\theta; x) - \gamma c(x, x_0)$, noting that the gradient steps is preformed as

$g^t = \nabla_\theta f(\theta^t, \hat{x}; x^t)$, where $\hat{x}$ is an approximate maximizer of $f(\theta, x; x^t)$ in $x$, and $\theta^{t+1} = \theta^t - \mu_t g^t$. We assume $\mu_t \leq \frac{1}{L_\psi}$ in the rest of the proof, which is satisfied for the constant step size $\mu = \sqrt{\frac{\Delta_\mathcal{L}}{L_\psi T \delta^2}}$ and $T \geq \frac{L_\psi \Delta_\mathcal{L}}{\delta^2}$. By a Taylor expansion using the $L_\psi$-smoothness of the objective $\mathcal{L}_k$ for $k$-th client, we have

$$\mathcal{L}_k\big|_{\theta^{t+1}} \leq \mathcal{L}_k\big|_{\theta^t} + \left\langle \nabla\mathcal{L}_k\big|_{\theta^t}, \theta^{t+1} - \theta^t \right\rangle + \frac{L_\psi}{2}||\theta^{t+1} - \theta^t||_2^2 \tag{10}$$

$$= \mathcal{L}_k\big|_{\theta^t} - \mu_t ||\nabla\mathcal{L}_k\big|_{\theta^t}||_2^2 + \frac{L_\psi \mu^2}{2}||g^t||_2^2 + \mu_t \left\langle \nabla\mathcal{L}_k\big|_{\theta^t}, \nabla\mathcal{L}_k\big|_{\theta^t} - g^t \right\rangle$$

$$= \mathcal{L}_k\big|_{\theta^t} - \mu_t \left(1 - \frac{1}{2}L_\psi\mu^2\right)||\nabla\mathcal{L}_k\big|_{\theta^t}||_2^2$$

$$+ \mu_t(1 - L_\psi\mu)\left\langle \nabla\mathcal{L}_k\big|_{\theta^t}, \nabla\mathcal{L}_k\big|_{\theta^t} - g^t \right\rangle + \frac{L_\psi\mu^2}{2}||g^t - \nabla\mathcal{L}_k\big|_{\theta^t}||_2^2$$

Consider the function $\phi_\gamma(\theta; x_0) = \sup_{x \in \mathbb{Z}} f(\theta, x; x_0)$, we define the potentially biased errors $\zeta^t = g^t - \nabla_\theta \phi_\gamma(\theta^t; x^t)$. Then we have the following relationship,

$$\mathcal{L}_k\big|_{\theta^{t+1}} \leq \mathcal{L}_k\big|_{\theta^t} - \mu_t \left(1 - \frac{1}{2}L_\psi\mu^2\right)||\nabla\mathcal{L}_k\big|_{\theta^t}||_2^2 \tag{11}$$

$$+ \mu_t(1 - L_\psi\mu)\left\langle \nabla\mathcal{L}_k\big|_{\theta^t}, \nabla\mathcal{L}_k\big|_{\theta^t} - \nabla_\theta\phi_\gamma(\theta^t; x^t) \right\rangle$$

$$- \mu_t(1 - L_\psi\mu_t)\left\langle \nabla\mathcal{L}_k\big|_{\theta^t}, \zeta^t \right\rangle + \frac{L_\psi\mu^2}{2}||\nabla_\theta\phi_\gamma(\theta^t; x^t) + \zeta^2 - \nabla\mathcal{L}_k\big|_{\theta^t}||_2^2$$

$$= \mathcal{L}_k\big|_{\theta^t} - \mu_t \left(1 - \frac{1}{2}L_\psi\mu^2\right)||\nabla\mathcal{L}_k\big|_{\theta^t}||_2^2$$

$$+ \mu_t(1 - L_\psi\mu)\left\langle \nabla\mathcal{L}_k\big|_{\theta^t}, \nabla\mathcal{L}_k\big|_{\theta^t} - \nabla_\theta\phi_\gamma(\theta^t; x^t) \right\rangle$$

$$- \mu_t(1 - L_\psi\mu_t)\left\langle \nabla\mathcal{L}_k\big|_{\theta^t}, \zeta^t \right\rangle$$

$$+ \frac{L_\psi\mu_t^2}{2}\left(||\zeta^t||_2^2 + ||\nabla_\theta\phi_\gamma(\theta^t; x^t) - \nabla\mathcal{L}_k\big|_{\theta^t}||_2^2 + 2\left\langle \nabla_\theta\phi_\gamma(\theta^t; x^t) - \nabla\mathcal{L}_k\big|_{\theta^t}, \zeta^t \right\rangle\right).$$

Since $\pm\langle a, b\rangle \leq \frac{1}{2}\left(||a||_2^2 + ||b||_2^2\right)$, we have

$$\mathcal{L}_k\big|_{\theta^{t+1}} \leq \mathcal{L}_k\big|_{\theta^t} - \frac{\mu_t}{2}||\nabla\mathcal{L}_k\big|_{\theta^t}||_2^2 + \mu_t((1 - L_\psi\alpha))\left\langle \nabla\mathcal{L}_k\big|_{\theta^t}, \nabla\mathcal{L}_k\big|_{\theta^t} - \nabla_\theta\phi_\gamma(\theta^t; x^t) \right\rangle \tag{12}$$

$$+ \frac{\mu_t((1 + L_\psi\mu))}{2}||\zeta||_2^2 + L_\psi\mu_t^2||\nabla_\theta\phi_\gamma(\theta^t; x^t) - \nabla\mathcal{L}_k\big|_{\theta^t}||_2^2.$$

Then, letting $x_*^t = \arg\max_x f(\theta^t, x; x^t)$, the error $\zeta^t$ satisfies,

$$||\zeta||_2^2 = ||\nabla_\theta\phi_\gamma(\theta^t; x^t) - \nabla f(\theta, \hat{x}^t; x^t)||_2^2 = ||\nabla_\theta\ell(\theta, x_*^t) - \nabla_\theta\ell(\theta, \hat{x}^t)||_2^2 \tag{13}$$

$$\leq L_{\theta_x}||\hat{x}^t - x_*^t||_2^2 \leq \frac{2L_{\theta_x}^2}{\lambda}\epsilon, \tag{14}$$

where the final inequality utilize the $\lambda = \gamma - L_{xx}$ strong-concavity of $x \mapsto f(\theta, x; x_0)$. For convenience, let $\hat{\epsilon} = \frac{2L_{\theta_x}^2}{\gamma - L_{xx}}\epsilon$. Taking conditional expectations in Eq. (12) and using $\mathbb{E}\left[\nabla_\theta\phi_\gamma(\theta^t; x^t)|\theta^t\right] = \nabla\mathcal{L}_k\big|_{\theta^t}$, we have,

$$\mathbb{E}\left[\mathcal{L}_k\big|_{\theta^{t+1}} - \mathcal{L}_k\big|_{\theta^t}|\theta^t\right] \leq -\frac{\mu_t}{2}||\nabla\mathcal{L}_k\big|_{\theta^t}||_2^2 + \frac{\mu_t((1 + L_\psi\mu))}{2}\hat{\epsilon} + L_\psi\mu_t^2||\nabla_\theta\phi_\gamma(\theta^t; x^t) - \nabla\mathcal{L}_k\big|_{\theta^t}||_2^2 \tag{15}$$

$$\leq -\frac{\mu_t}{2}||\nabla\mathcal{L}_k\big|_{\theta^t}||_2^2 + \mu_t\hat{\epsilon} + L_\psi\mu_t^2||\nabla_\theta\phi_\gamma(\theta^t; x^t) - \nabla\mathcal{L}_k\big|_{\theta^t}||_2^2$$

Since $\mu_t \leq \frac{1}{L_\psi}$, taking a fixed step size $\mu$, we have

$$\mathbb{E}\left[||\nabla\mathcal{L}_k\big|_{\theta^t}||_2^2\right] - 2\hat{\epsilon} \leq \frac{2}{\mu}\mathbb{E}\left[\mathcal{L}_k\big|_{\theta^t} - \mathcal{L}_k\big|_{\theta^{t+1}}\right] + 2L_\psi\mu\delta^2 \tag{16}$$

Because $\mathbb{E}\left[||\nabla_\theta \phi_\gamma(\theta; Z) - \nabla\mathcal{L}_k|_{(\theta)}||_2^2\right] \leq \delta^2$, summing over $t$, we have,

$$\frac{1}{T}\sum_t^T \mathbb{E}\left[||\nabla\mathcal{L}_k|_{\theta^t}||_2^2\right] - 2\hat{\epsilon} \leq \frac{2}{\mu T}(\mathcal{L}_k|_{\theta^0} - \mathbb{E}[\mathcal{L}_k|_{\theta^T}]) + 2L_\psi\mu\delta^2$$

$$\leq \frac{2\Delta}{\mu T} + 2L_\psi\mu\delta^2 \qquad (17)$$

Since $\mu = \sqrt{\frac{Delta}{L_\psi T\delta^2}}$, and $\lambda = \gamma - L_{xx}$, we can get the following result,

$$\frac{1}{T}\sum_t^T \mathbb{E}\left[||\nabla\mathcal{L}_k|_{\theta^t}||_2^2\right] \leq \frac{4L_{\theta x}^2\epsilon}{\lambda} + 4\delta\sqrt{\frac{L\Delta}{T}} \qquad (18)$$

Adopting our $\alpha$-weighted relaxation, we have,

$$\frac{1}{T}\sum_t^T \mathbb{E}\left[\left|\left|\nabla\mathcal{L}^\alpha(\widehat{K})|_{\theta^t}\right|\right|_2^2\right]$$

$$= \frac{1}{T}\sum_t^T \mathbb{E}\left[\left|\left|(1+\alpha)\sum_{k=1}^{\widehat{K}}\frac{N_{\phi(k)}^{(t)}}{N}\nabla\mathcal{L}_{\phi(k)}|_{\theta^t} + (1-\alpha)\sum_{k=\widehat{K}+1}^{K}\frac{N_{\phi(k)}^{(t)}}{N}\nabla\mathcal{L}_{\phi(k)}|_{\theta^t}\right|\right|_2^2\right]$$

$$\leq \frac{1}{T}\sum_t^T \left((1+\alpha)\sum_{k=1}^{\widehat{K}}\frac{N_{\phi(k)}^{(t)}}{N}\mathbb{E}\left[\left|\left|\nabla\mathcal{L}_{\phi(k)}|_{\theta^t}\right|\right|_2^2\right] + (1-\alpha)\sum_{k=\widehat{K}+1}^{K}\frac{N_{\phi(k)}^{(t)}}{N}\mathbb{E}\left[\left|\left|\nabla\mathcal{L}_{\phi(k)}|_{\theta^t}\right|\right|_2^2\right]\right)$$

$$\leq \frac{1}{T}\sum_t^T \left((1+\alpha)\sum_{k=1}^{\widehat{K}}\frac{N_{\phi(k)}^{(t)}}{N} + (1-\alpha)\sum_{k=\widehat{K}+1}^{K}\frac{N_{\phi(k)}^{(t)}}{N}\right)\left(\frac{4L_{\theta x}^2\epsilon}{\lambda} + 4\delta\sqrt{\frac{L\Delta}{T}}\right)$$

$$= \frac{1}{T}\sum_t^T \left(1 + \alpha\frac{\sum_{k=1}^{\widehat{K}} N_{\phi(k)}^{(t)} - \sum_{k=\widehat{K}+1}^{K} N_{\phi(k)}^{(t)}}{N}\right)\left(\frac{4L_{\theta x}^2\epsilon}{\lambda} + 4\delta\sqrt{\frac{L\Delta}{T}}\right)$$

$$= \left(1 + \alpha\frac{\frac{1}{T}\sum_t^T \xi^{(t)}}{N}\right)\left(\frac{4L_{\theta x}^2\epsilon}{\lambda} + 4\delta\sqrt{\frac{L\Delta}{T}}\right),$$

$$(19)$$

where $\xi^{(t)} = \sum_{k=1}^{\widehat{K}} N_{\phi(k)}^{(t)} - \sum_{k=\widehat{K}+1}^{K} N_{\phi(k)}^{(t)}$ to simplify the notations. $\qquad\square$

## C  PROOF OF THEOREM 4.3

Based on the convergence of FedAvg (Li et al., 2019), we proof Theorem 4.3 in this section.

First, we make the following assumptions and present some useful lemmas. Specifically, we make the following assumptions. Assumption C.1 and C.2 are standard (typical examples are the $\ell_2$-norm regularized linear regression, logistic regression, or softmax classifier). Assumption C.3 and C.4 have been made by the previous works (Zhang et al., 2013; Li et al., 2019).

**Assumption C.1.** $\mathcal{L}_1, \ldots, \mathcal{L}_K$ are all $L$-smooth: for all $v$ and $w$, $\mathcal{L}_k(v) \leq \mathcal{L}_k(w) + (v - w)^T\nabla\mathcal{L}_k(w) + \frac{L}{2}||v - w||_2^2$.

**Assumption C.2.** $\mathcal{L}_1, \ldots, \mathcal{L}_K$ are all $\lambda$-strongly convex: for all $v$ and $w$, $\mathcal{L}_k(v) \geq \mathcal{L}_k(w) + (v - w)^T\nabla\mathcal{L}_k(w) + \frac{\lambda}{2}||v - w||_2^2$.

**Assumption C.3.** Let $\xi_t^k$ be sampled from the $k$-th device's local data uniformly at random. The variance of stochastic gradients in each device is bounded: $\mathbb{E}||\nabla\mathcal{L}_k(w_t^k, \xi_t^k) - \nabla\mathcal{L}_k(w_t^k)||^2 \leq \delta_k^2$ for $k = 1, \cdots, K$.

**Assumption C.4.** The expected squared norm of stochastic gradients is uniformly bounded, i.e., $\mathbb{E}||\nabla\mathcal{L}_k(w_t^k, \xi_t^k)||^2 \leq \varsigma^2$ for all $k = 1, \cdots, K$ and $t = 1, \cdots, T - 1$.

We use the following lemmas proved by Li et al. (2019). Let $\theta_t^k$ denotes the model parameter maintained in the $k$-th client at $t$-th step, $\Theta$ represents an immediate result of one step SGD update from $\theta_t^k$. For convenience, we define $\bar{\Theta}_t = \sum_{k=1}^K \frac{N_k}{N}\Theta_t$, $\bar{\theta}_t == \sum_{k=1}^K \frac{N_k}{N}\theta_t$, $\bar{g}_t = \sum_{k=1}^K \frac{N_k}{N}\nabla\mathcal{L}_k(\theta_t^k)$ and $g_t = \sum_{k=1}^K \frac{N_k}{N}\nabla\mathcal{L}_k(\theta_t^k, \xi_t^k)$. Therefore, $\mathbb{E}g_t = \bar{g}_t$.

**Lemma C.1** (Results of one step SGD). *Assume Assumption C.1 and C.2. If $\eta_t \leq \frac{1}{4L}$, we have*

$$\mathbb{E}||\bar{\Theta}_{t+1} - \theta^*||^2 \leq (1 - \eta_t\lambda)\mathbb{E}||\bar{\theta}_t - \theta^*||^2 \tag{20}$$

$$+ \eta_t^2 \mathbb{E}||g_t - \bar{g}_t||^2 + 6L\eta_t^2\Gamma + 2\mathbb{E}\sum_{k=1}^K \frac{N_k}{N}||\bar{\theta}_t - \theta_t^k||^2,$$

*where $\Gamma = \mathcal{L}^* - \sum_k^K \mathbb{E}_t \frac{N_k}{N}\mathcal{L}_k^* \geq 0$*

**Lemma C.2** (Bounding the variance). *Assume Assumption C.3. It follows that*

$$\mathbb{E}||g_t - \bar{g}_t||^2 \leq \sum_{k=1}^K \left(\frac{N_k}{N}\right)^2 \delta_k^2, \tag{21}$$

**Lemma C.3** (Bounding the divergence of $\theta_t^k$). *Assume Assumption C.4, that $\eta_t$ is non-increasing and $\eta \leq 2\eta_{t+E}$ for all $t > 0$. It follows that*

$$\mathbb{E}\left[\sum_{k=1}^K \frac{N_k}{N}||\bar{\theta}_t - \theta_t^k||^2\right] \leq 4\eta_t^2(E-1)^2\varsigma^2 \tag{22}$$

*proof of Theorem 4.3.* Let $\Delta_t = \mathbb{E}||\theta_t - \theta^*||^2$. From Lemma C.1, Lemma C.2 and Lemma C.3, it follows that

$$\Delta_{t+1} \leq (1 - \eta_t\lambda)\Delta_t + \eta_t^2 B, \tag{23}$$

where,

$$B = \sum_k^K \left(\frac{P_k^{(T)}}{1 + \alpha\frac{\xi^{(T)}}{N}}\right)^2 \left(\frac{N_k}{N}\delta_k\right)^2 + 6L\left(\mathcal{L}^* - \sum_k^K \frac{P_k^{(T)}}{1 + \alpha\frac{\xi^{(T)}}{N}}\frac{N_k}{N}\mathcal{L}_k^*\right) + 8(E-1)^2\varsigma^2, \tag{24}$$

and $\left(\frac{P_k^{(T)}}{1+\alpha\frac{\xi^{(T)}}{N}}\right)^2$ and $\frac{P_k^{(T)}}{1+\alpha\frac{\xi^{(T)}}{N}}$, are acted as the scalar timing by the personalized variance bound $\delta_k^2$ and the local optimum $\mathcal{L}_k^*$ of each client.

For a diminishing stepsize, $\eta_t = \frac{\beta}{\gamma+t}$ for some $\beta > \frac{1}{\lambda}$ and $\gamma > 0$ such that $\eta_1 \leq \min\{\frac{1}{\lambda}, \frac{1}{4L}\} = \frac{1}{4L}$ and $\eta_t \leq 2\eta_{t+E}$. We will prove that $\Delta_t \leq \frac{\nu}{\gamma+t}$, where $\nu = \max\{\frac{\beta^2 B}{\beta\lambda-1}, (\gamma+1)\Delta_1\}$. The above can be proved by induction. Firstly, the definition of $\nu$ ensures that it holds for $t = 1$. Assume the conclusion holds for some $t$, it follows that,

$$\begin{aligned}
\Delta_{t+1} &\leq (1 - \eta_t\lambda)\Delta_t + \eta_t^2 B \\
&\leq (1 - \frac{\beta\lambda}{t+\gamma})\frac{\nu}{t+\gamma} + \frac{\beta^2 B}{(t+\gamma)^2} \\
&= \frac{t+\gamma-1}{(t+\gamma)^2}\nu + [\frac{\beta^2 B}{(t+\gamma)^2} - \frac{\beta\lambda-1}{(t+\gamma)^2}\nu] \\
&\leq \frac{\nu}{t+\gamma+1}.
\end{aligned} \tag{25}$$

Then by the L-smoothness of $\mathcal{L}(\cdot)$,

$$\mathbb{E}[\mathcal{L}_t] - \mathcal{L}^* \leq \frac{L}{2}\Delta_t \leq \frac{L}{2}\frac{\nu}{\gamma+t} \tag{26}$$

Specifically, if we choose $\beta = \frac{2}{\lambda}, \gamma = \max\{8\frac{L}{\lambda}, E\} - 1$ and denote $\kappa = \frac{L}{\lambda}$, then $\eta_t = \frac{2}{\lambda}\frac{1}{\gamma+t}$. One can verify that the choice of $\eta_t$ satisfies $\eta_t \leq 2\eta_{t+E}$ for $t \geq 1$. Then we have

$$\nu = \max\left\{\frac{\beta^2 B}{\beta\lambda-1}, (\gamma+1)\Delta_1\right\} \leq \frac{\beta^2 B}{\beta\lambda-1} + (\gamma+1)\Delta_1 \leq \frac{4B}{\lambda^2} + (\gamma+1)\Delta_1, \tag{27}$$

and

$$\mathbb{E}[\mathcal{L}_t] - \mathcal{L}^* \leq \frac{L}{2}\frac{\nu}{\gamma+t} \leq \frac{\kappa}{\gamma+t}\left(\frac{2B}{\lambda} + \frac{\lambda(\gamma+1)}{2}\Delta_1\right) \tag{28}$$

$\square$

## D  LEARNING FRAMEWORK AND ALGORITHM

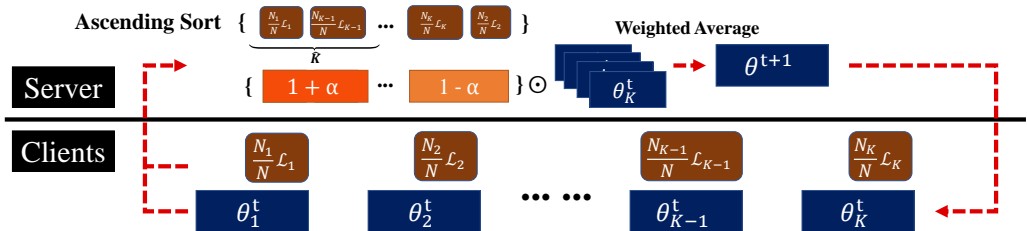

Figure 5: A brief illustration of our $\alpha$-Weighted Federated Adversarial Training ($\alpha$-WFAT) framework. On the client-side, each client will conduct adversarial training with its local data and update the optimized model parameter (*i.e.,* $\theta_k$) with the adversarial loss (*i.e.,* $\frac{N_k}{N}\mathcal{L}_k$)). On the server-side, after collecting the model parameters and the loss value (information of the robustness), the server will conduct an ascending sort and aggregate the global model with a weighted average (denoted by $\odot$) which upweights the top populations of the robust client's model parameters with $\alpha$.

Here, we provide an intuitive illustration of our proposed $\alpha$-WFAT in Figure 5. Based on the $\alpha$-weighted mechanism, we provide a new flexible framework for the combination of adversarial training with federated learning. It is orthogonal to a variety of different adversarial training (Zhang et al., 2019; Wu et al., 2020; Zhang et al., 2020; Chen et al., 2020; 2021; Wang et al., 2020a; Jiang et al., 2020; Wang et al., 2020c) methods and federated learning algorithms (Li et al., 2018; Kairouz et al., 2019; Hong et al., 2021) which gain the adversarial robustness or alleviate the data heterogeneity on the client side, and can be simply but effectively combined with other approach.

## E  EXPERIMENTAL DETAILS

**Dataset.**  We conduct the experiments on three benchmark datasets, *i.e., SVHN* (Netzer et al., 2011), *CIFAR-10* and *CIFAR-100* (Krizhevsky, 2009) for federated adversarial training. For the IID scenario, we randomly distribute these datasets to each client. For simulating the Non-IID scenario, we follow McMahan et al. (2017); Shah et al. (2021) to distribute the training data based on their labels. To be specific, a skew parameter $s$ is utilized in the data partition introduced by Shah et al. (2021), which enables $K$ clients to get a majority of the data samples from a subset of classes. We denote the set of all classes in a dataset as $\mathcal{Y}$ and create $\mathcal{Y}_k$ by dividing all the class labels equally among $K$ clients. Accordingly, we split the data across $K$ clients that each client has $(100 - (K-1) \times s)\%$ of data for the class in $\mathcal{Y}_k$ and $s\%$ of data in other split sets. In our experiments, we set $s = 2$ for simulating the Non-IID partition with 5 clients as Shah et al. (2021) recommended.

**Training and Evaluation.**  In the experiments, we follow the previous works to leverage the same architectures, *i.e., NIN* (Lin et al., 2014) for *CIFAR-10*, *ResNet-18* (He et al., 2016) for *CIFAR-100* and *Small CNN* (Zhang et al., 2020) for *SVHN*. For the local training batch size, we set 32 for *CIFAR-10*, 128 for *CIFAR-100* and *SVHN*. For the training schedule, SGD is adopted with 0.9 momentum for 100 communication rounds under 5 clients as in (Hong et al., 2021; Shah et al., 2021), and the weight decay = 0.0001. For adversarial training, we set the configurations of PGD respectively (Madry et al., 2018) for different datasets. On *CIFAR-10/CIFAR-100*, we set the perturbation bound $\epsilon = 8/255$, the PGD step size $2/255$ and set the PGD step number 10. On *SVHN*, we set the perturbation bound $\epsilon = 4/255$, the PGD step size $1/255$, and keep the same step number 10. Regarding the evaluation, the accuracy for the natural test data and that for the adversarial test data are computed

Table 3: Brief summary of the experimental details about $\alpha$-WFAT

| Dataset | Network | $K$ | $\hat{K}$ | $\alpha$ |
|---------|---------|-----|-----------|----------|
| *CIFAR-10* | NIN (Shah et al., 2021) | 5 | 1 | 1/6 |
| *CIFAR-100* | ResNet-18 (Chen et al., 2021) | 5 | 1 | 1/41 |
| *SVHN* | SmallCNN (Zhang et al., 2019) | 5 | 1 | 1/11 |

following Wang et al. (2019); Wu et al. (2020). Note that, the adversarial test data are generated by FGSM, PGD-20, C&W (Carlini & Wagner, 2017) attack with the same perturbation bound and step size as the training. All the adversarial generations have a random start, i.e, the uniformly random perturbation of $[-\epsilon, \epsilon]$ added to the natural data before attacking iterations. Besides, we also report the robustness under a stronger AutoAttack, termed as AA for simplicity.

As for our $\alpha$-WFAT, different training tasks adopt different $\alpha$-weighted mechanism considering different characteristic of local training data, we set $\alpha = 1/6$ (*i.e.,* $\frac{1+\alpha}{1-\alpha} = 1.4$) for the experiments on *CIFAR-10*, and $\alpha = 1/41$ (*i.e.,* $\frac{1+\alpha}{1-\alpha} = 1.05$) for the experiments on *CIFAR-100* and $\alpha = 1/11$ (*i.e.,* $\frac{1+\alpha}{1-\alpha} = 1.2$) for the experiments on *SVHN*. As for FedProx, we set $\mu = 0.01$ for each dataset and its $\alpha$ for $\alpha$-weighted mechanism are $1/11$, $1/41$, and $1/11$. As for Scaffold, the $\alpha$ adopted for previous datasets are $1/11$, $1/101$ and $1/11$ respectively.

As for the choice of the hyper-parameter $\alpha$, one useful way to set it might be progressively probing its effect in a value-growth manner. When the $\alpha$ is very small, the objective will approximately degenerate the original objective of FAT, so does the performance with no harm. Slightly enlarging $\alpha$ can improve the performance due to the benefit of the bias-variance trade-off, and then make a stop in one point where the performance becomes drop.

### E.1 EMPHASIZE/DE-EMPHASIZE IN OUR $\alpha$-WFAT

We conduct an empirical comparison between $\alpha$-WFAT that emphasizes (relatively de-emphasize the other clients) the client model with the smallest loss and a contrary variant that de-emphasizes it (relatively emphasize the other clients) as follows. We find that de-emphasizing the client with smallest loss (relatively emphasize those with larger loss) consistently harm the model performance across these evaluations.

Table 4: Comparison with emphasize/de-emphasize the client with smallest loss.

| Setting | | | Non-IID | | | |
|---------|---|---|---------|---|---|---|
| CIFAR-10 | | | Natural | FGSM | PGD-20 | $CW_\infty$ |
| FedAvg | $\alpha$-**WFAT**: $\frac{1+\alpha}{1-\alpha} = 1.4$ | emphasize | **63.44%** | **45.13%** | **37.17%** | **33.99%** |
| | $\alpha$-**WFAT**: $\frac{1+\alpha}{1-\alpha} = 1.2$ | emphasize | 62.26% | 44.08% | 35.83% | 33.31% |
| | FAT: $\frac{1+\alpha}{1-\alpha} = 1.0$ | original | 57.45% | 39.44% | 32.58% | 30.52% |
| | $\alpha$-**WFAT**: $\frac{1+\alpha}{1-\alpha} = 0.8$ | de-emphasize | 50.45% | 34.34% | 27.86% | 26.62% |
| | $\alpha$-**WFAT**: $\frac{1+\alpha}{1-\alpha} = 0.6$ | de-emphasize | 40.47% | 28.81% | 24.36% | 23.19% |
| SVHN | | | Natural | FGSM | PGD-20 | $CW_\infty$ |
| FedAvg | $\alpha$-**WFAT**: $\frac{1+\alpha}{1-\alpha} = 1.4$ | emphasize | 90.60% | 87.75% | **73.12%** | **70.51%** |
| | $\alpha$-**WFAT**: $\frac{1+\alpha}{1-\alpha} = 1.2$ | emphasize | **91.25%** | **88.28%** | 71.72% | 69.79% |
| | FAT: $\frac{1+\alpha}{1-\alpha} = 1.0$ | original | 91.24% | 87.95% | 68.87% | 67.89% |
| | $\alpha$-**WFAT**: $\frac{1+\alpha}{1-\alpha} = 0.8$ | de-emphasize | 90.03% | 86.12% | 64.35% | 64.32% |
| | $\alpha$-**WFAT**: $\frac{1+\alpha}{1-\alpha} = 0.6$ | de-emphasize | 89.46% | 84.80% | 58.64% | 58.96% |

### E.2 MEAN AND STD RESULTS OF THE NON-IID SETTINGS

Considering that the Non-IID results are sensitive to the selection of data in each client, we conduct our experiments on Non-IID settings for multiple times and conclude the results as follows. In summary, our $\alpha$-WFAT can consistently improve the model robustness with comparable or even better natural performance than previous federated optimization methods.

Table 5: Performance on Non-IID settings under different federated optimization methods (Mean±Std).

| Setting | | Non-IID | | | | |
|---|---|---|---|---|---|---|
| CIFAR-10 | | Natural | FGSM | PGD-20 | $CW_\infty$ | AA |
| FedAvg | FAT | 58.13±0.68% | 40.06±0.62% | 32.56±0.01% | 30.88±0.37% | 29.17±0.03% |
| | $\alpha$-**WFAT** | **63.36±0.07%** | **44.82±0.32%** | **37.14±0.03%** | **33.39±0.61%** | **31.66±0.70%** |
| FedProx | FAT | 59.95±0.45% | 41.44±0.15% | 33.83±0.01% | 31.65±0.36% | 30.11±0.09% |
| | $\alpha$-**WFAT** | **62.04±0.47%** | **44.21±0.08%** | **36.64±0.11%** | **32.62±0.20%** | **31.83±0.15%** |
| Scaffold | FAT | 61.44±1.37% | 42.85±0.76% | 34.08±0.05% | 32.56±0.02% | 31.03±0.08% |
| | $\alpha$-**WFAT** | **63.16±0.96%** | **45.55±0.50%** | **37.33±0.02%** | **34.82±0.04%** | **33.32±0.01%** |
| CIFAR-100 | | Natural | FGSM | PGD-20 | $CW_\infty$ | AA |
| FedAvg | FAT | 31.93±0.85% | 20.04±0.44% | 16.34±0.18% | 13.65±0.28% | 12.70±0.03% |
| | $\alpha$-**WFAT** | **34.80±0.37%** | **20.91±0.35%** | **16.66±0.06%** | **13.78±0.13%** | **12.79±0.04%** |
| FedProx | FAT | **34.07±0.74%** | 20.49±0.29% | 16.20±0.12% | 13.68±0.06% | 12.67±0.08% |
| | $\alpha$-**WFAT** | 33.95±0.39% | **20.86±0.40%** | **16.73±0.01%** | **13.80±0.04%** | **12.80±0.02%** |
| Scaffold | FAT | **39.89±1.28%** | 24.78±0.40% | 19.82±0.20% | 16.73±0.01% | 15.51±0.02% |
| | $\alpha$-**WFAT** | 39.80±1.27% | **25.05±0.51%** | **20.27±0.13%** | **16.79±0.01%** | **15.58±0.01%** |
| SVHN | | Natural | FGSM | PGD-20 | $CW_\infty$ | AA |
| FedAvg | FAT | **91.52±0.28%** | 88.13±0.18% | 68.98±0.11% | 68.04±0.15% | 66.59±0.04% |
| | $\alpha$-**WFAT** | 91.26±0.01% | **88.27±0.02%** | **72.04±0.32%** | **69.96±0.16%** | **68.89±0.27%** |
| FedProx | FAT | 91.00±0.08% | 87.65±0.15% | 68.48±0.04% | 67.16±0.02% | 65.76±0.18% |
| | $\alpha$-**WFAT** | **91.19±0.06%** | **88.15±0.01%** | **71.84±0.30%** | **69.88±0.35%** | **68.84±0.37%** |
| Scaffold | FAT | 90.82±0.87% | 87.89±0.66% | 69.51±0.84% | 68.12±0.88% | 67.19±0.54% |
| | $\alpha$-**WFAT** | **90.93±0.76%** | **88.27±0.45%** | 71.77±0.38% | 69.49±0.67% | **68.37±0.48%** |

