# OpenReview forum: "$\alpha$-Weighted Federated Adversarial Training"
_ICLR.cc/2022/Conference — ICLR 2022 Submitted_

### Official Review · Reviewer_m6tB · 2021-10-20

**Correctness:** 3
**Technical Novelty And Significance:** 2
**Empirical Novelty And Significance:** 2
**Recommendation:** 5
**Confidence:** 4

**Main Review:**

The paper tackles an interesting direction in federated learning, and is well-organized. Overall, I enjoyed the paper itself and the presentation of ideas, but I do think the contribution is incremental. I listed the detailed questions and suggestions below.

1. What is the detailed settting for Fig 1? What is the AT method used for training, PGD-AT or TRADES? Is the dataset here non-IID or IID? PGD-20 or PGD-10?
2. The novelty of this paper is incremental. It adopted the weighting strategy. I am wondering whether the global model can be trained well in such a way.
3. Many federated learning settings have much larger client numbers (thousands or even millions, only 5 clients in this paper, ), and it is unclear if the proposed solution scales.  Besides, only a slight improvement can be seen in Tabel 2.
4. This method doesn't make sense to me, especially for Fig 2. The motivation for the specific weighting chosen was not very clearly explained.



**Summary Of The Paper:**

The authors explore the adversarial robustness of federated learning. They claim that the inner-maximization optimization of AT can exacerbate the data heterogeneity among local clients. They propose an algorithm, $\alpha$--WFAT, which relaxes the inner-maximization of Adversarial Training into a lower bound friendly to Federated Learning.. The authors also experimentally establish that federated learning models are most susceptible to attacks when clients are using non-IID training sets. The experiments are performed over the CIFAR-10 , SVHN and CIFAR-100datasets.

**Summary Of The Review:**

The paper tackles an interesting direction in federated learning, and is well-organized. Overall, I enjoyed the paper itself and the presentation of ideas, but I do think the contribution is incremental.

---

> ### Author Response · Authors · 2021-11-21
> **Response to Reviewer m6tB: Part 1**
>
> > Q1: What is the detailed settting for Fig 1? What is the AT method used for training, PGD-AT or TRADES? Is the dataset here non-IID or IID? PGD-20 or PGD-10?
>
> **A1:** The AT method used for training in Fig 1 is PGD-AT, and the dataset used for Fig 1 is non-IID. As for robust evaluation, we adopted PGD-20 here to check the model performance. Thank you for the question and we will improve the caption of Fig 1 to make the related setting clear.
>
> > Q2: The novelty of this paper is incremental. It adopted the weighting strategy. I am wondering whether the global model can be trained well in such a way.
>
> **A2:** We would like to defend that our novelty is not incremental, given the only most related work Federated Adversarial Training [1]. Before our work, no one points out why the combination of adversarial training and federated learning leads to the performance drop. We trace back that the inner maximization exacerbates the heterogeneity issue and might induce the global model to converge to the undesired local optimal. Besides, the weighting strategy is under-explored and not off-the-shelf, since we do not know how to upweight or downweight what kinds of the client models. We start from a lower bound relaxation and provide the corresponding theoretical analysis on the bias-variance trade-off. Then, we extend FAT with the aid of the lower bound relaxation to propose our $\alpha$-WFAT. Simultaneously, we give the corresponding theoretical asymptotics to the optimal point. The extensive experiments demonstrate that $\alpha$-WFAT consistently outperforms FAT and is compatible with a range of Federated optimization methods and adversarial training paradigms. We believe our work is simple, effective, and novel enough in terms of the current area development.
>
> Regarding the concern about the training of $\alpha$-WFAT, except the experiments shown in the submission, we have also provided the comprehensive ablation study in Section 5.2 as well as training and evaluation details in Appendix, and will upload our code and welcome the peers to reproduce our experiments.
>
> [1] Zizzo, Giulio, et al. "Fat: Federated adversarial training." In NeurIPS workshop. 2020.

---

> ### Author Response · Authors · 2021-11-21
> **Response to Reviewer m6tB: Part 2**
>
> > Q3: Many federated learning settings have much larger client numbers (thousands or even millions, only 5 clients in this paper, ), and it is unclear if the proposed solution scales. Besides, only a slight improvement can be seen in Tabel 2.
>
> **A3:** For the training settings, we follow the previous related studies [1,2] to consider appropriate client numbers for federated adversarial training in those benchmarked datasets. In addition, considering adversarial training, the training sizes of the benchmarked datasets like CIFAR-10, CIFAR-100 and SVHN are 50000, 50000 and 73257. The sizes of local training data in each client might be not enough to support such difficult adversarial training tasks (e.g., each client only has 1-10 training examples if using thousands or even millions of clients). Hence, we mainly adopted 5 clients in our performance evaluation part. And we also adopted 8 clients in Fig 4 to explore the effects of $\widehat{K}/K$. For more clients, our method can be adapted by adjusting the proportion of clients whose weights need to be changed. However, we would like to span our explore if there are benchmarked large datasets which are appropriate to apply and research adversarial training under federated settings and can simultaneously support thousands or even millions of clients in this task.
>
> Besides, for the experiments in Table 2, we believe the improvement is not incremental. For example, in CIFAR-10, our $\alpha$-WFAT can significantly outperform FAT on the Non-IID data in terms of both the natural accuracy (about 2%-6%) and the robust accuracy (about 2%-5%) according to those evaluation metrics. This kind of enhancement is significant, since the FedProx also has a similar or even less improvement on the basis of FedAvg (compared with our performance enhancement on FedAvg), and we can further improve the model robustness on the basis of those federated optimization methods. And we have also reported the results in the Mean$\pm$Std formula in Q3 of Reviewer 4Hkz to exclude the random effects (the results also will be updated into our Appendix), and it again confirmed that our proposed method can consistently improve the robust performance with comparable or even better natural performance.
>
> [1] Zizzo, Giulio, et al. "Fat: Federated adversarial training." In NeurIPS workshop. 2020.
>
> [2] Shah, Devansh, et al. "Adversarial training in communication constrained federated learning." In arXiv. 2021.
>
> > Q4: This method doesn't make sense to me, especially for Fig 2. The motivation for the specific weighting chosen was not very clearly explained.
>
> **A4:** We have greatly improved the description of Figure 2 and the Motivation section, and kindly refer the reviewer to the updated submission. In summary, we use Figure 2 to characterize the client model with the larger loss (more adversarial examples close to the decision boundary) is not good for robustness as a similar validation in [1]. Then the proper relaxation like Eq. (3) to selectively emphasize/de-emphasize the client models will help aggregate a better global model. Regarding to the weighting strategy, we kindly refer to the A3 to Q3 of Reviewer 4ioa.
>
> [1] Hitaj, Dorjan, et al. "Evaluating the Robustness of Geometry-Aware Instance-Reweighted Adversarial Training." In arXiv. 2021.

---

> ### Author Response · Authors · 2021-11-22
> **Need further clarification?**
>
> Thanks very much for your constructive comments on our work. We have tried our best to address the concerns. Is there any unclear point so that we should/could further clarify?

---

> ### Author Response · Authors · 2021-11-26
> **Would you mind confirming if you have further questions? Thanks!**
>
> Dear Reviewer m6tB,
>
> As approach the end of the discussion period, we highly appreciate knowing if our responses have addressed your initial questions. We are delighted to answer your remaining concern. We appreciate your inputs and feedback very much. Thank you!
>
> Best wishes,
> The authors of Paper2033

---

### Official Review · Reviewer_4ioa · 2021-10-29

**Correctness:** 2
**Technical Novelty And Significance:** 3
**Empirical Novelty And Significance:** 4
**Recommendation:** 5
**Confidence:** 3

**Main Review:**

Despite reasonable performance improvement, I am not convinced by many of the claims made by this paper:
It claims the biased adversarial generation in the local machine causes the issue in FAT. But the adversarial x tilde is purely determined by the current estimation of theta, and during the training, no local machine knows the optimal theta. So technically, adversarial samples are always biased, even for centralized adversarial training.

In the discussion of figure 2, I don't understand why hard-to-train local machines may "over-optimize" their model. Does the "over-optimize" mean "overfit" the local data? If so, then the whole argument applies to federated standard training as well, and the alpha-weighted mechanism shall improve federated standard training. But the authors clearly say that the alpha-weighted mechanism doesn't work for federated standard training.

On the other hand, does hard-to-train local data contain more valuable information of the real decision boundary, since data is close to the boundary? Therefore, they should be emphasized rather than de-emphasized.

Several notations are unclear, for example
L-smoothness: l(.,.) is L-smoothness. Do you mean l(f_theta(.),.) is L-smooth w.r.t. theta uniformly for all tilde x and y?
Thm 4.2: E|\nabla L^\alpha(\hat K)-\nabla_\theta(f(\tilde x),y)|^2, what's the tilde x and y, one random sample?
Thm 4.3: E|\theta^0-\theta^*|^2 term. What is the meaning of this expectation?

**Summary Of The Paper:**

This paper introduces the alpha Weighted Federated Adversarial Training algorithm. The key of the idea is that in the aggregate step, the center prefers the local machine that yields smaller lost. Some theoretical results are delivered with numerical experiments. The paper claims that the alpha-weighted mechanism is tailored for the inner-maximization of Federated Adversarial Training, which is the rationale of the whole work.

**Summary Of The Review:**

Empirical results are good, but the rationale of the proposed method is unclear and not well explained. The theoretical results don't justify the method, instead, within the proposed context, justify algorithmic convergence.

---

> ### Author Response · Authors · 2021-11-21
> **Response to Reviewer 4ioa: Part1**
>
> > Q1: Despite reasonable performance improvement, I am not convinced by many of the claims made by this paper: It claims the biased adversarial generation in the local machine causes the issue in FAT. But the adversarial x tilde is purely determined by the current estimation of theta, and during the training, no local machine knows the optimal theta. So technically, adversarial samples are always biased, even for centralized adversarial training.
>
> **A1:** We agree that the adversarial examples might be biased in the centralized adversarial training, where the corresponding issue noticed by the recent study is robustness overfitting [1]. In terms of the combination of adversarial training and federated learning, the key point is that such adversarial examples exacerbate the heterogeneity, which induces the performance deterioration under the model aggregation methods *e.g.,* FedAvg, FedProx, and Scaffold. One rough explanation is that it makes the model hard to converge to a globally optimal, and we introduce a biased objective relaxation to accelerate the convergence as analyzed by Theorem 4.2, *i.e.,* the bias-variance trade-off [2]. We will improve the description about the Motivation section to clearly emphasize the bias-variance trade-off and connect to our proposal Eq.(3).
>
> [1]  Rice, Leslie, Eric Wong, and Zico Kolter. "Overfitting in adversarially robust deep learning." In ICML. 2020.
>
> [2] Kohavi, Ron, and David H. Wolpert. "Bias plus variance decomposition for zero-one loss functions." In ICML. 1996.
>
> > Q2: In the discussion of figure 2, I don't understand why hard-to-train local machines may "over-optimize" their model. Does the "over-optimize" mean "overfit" the local data? If so, then the whole argument applies to federated standard training as well, and the alpha-weighted mechanism shall improve federated standard training. But the authors clearly say that the alpha-weighted mechanism doesn't work for federated standard training.
>
> **A2:** We apologize that the inexact description confuses the reviewer. The "hard-to-train" local machines mean the client model with the larger loss, and "over-optimize" means that the "hard-to-train" local machines might make a dominating effect on the convergence of the global model, given the current model aggregation methods treat all client models indiscriminately. We use Figure 2 and the motivation section to tell that it might be better to selectively weight the client models in the model aggregation as the relaxation in Eq. (3). Please see the improved Motivation section.

---

> ### Author Response · Authors · 2021-11-21
> **Response to Reviewer 4ioa: Part2**
>
> > Q3: On the other hand, does hard-to-train local data contain more valuable information of the real decision boundary, since data is close to the boundary? Therefore, they should be emphasized rather than de-emphasized.
>
> **A3:** In the case of federated adversarial training, the enlarged heterogeneity usually means that we over-emphasize the hard-to-train data that is close to the decision boundary. The convergence of the global model might be pushed to the undesired local optimal by the client models with the larger losses. A validation that supports this point in the area of adversarial training is [1], which finds emphasizing the data close to the decision boundary is not good for robustness. To verify the fact, we conduct an empirical comparison between our $\alpha$-WFAT that emphasizes (relatively de-emphasize the other clients) the client model with the smallest loss and a contrary variant that de-emphasizes it (relatively emphasize the other clients) as follows (In Table S1). We find that de-emphasizing the client with the smallest loss (relatively emphasize those with larger loss) consistently harms the model performance.
>
> **Table S1.** Comparison with emphasizing/de-emphasizing the client with smallest adversarial loss.
>
> |        |             **CIFAR-10 (Non-IID)**             |              | **Natural** |  **FGSM**  | **PGD-20** |   **CW**   |
> | :----: | :--------------------------------------------: | :----------: | :---------: | :--------: | :--------: | :--------: |
> | FedAvg | $\alpha$-WFAT: $\frac{1+\alpha}{1-\alpha}=1.4$ |  emphasize   | **63.44%**  | **45.13%** | **37.17%** | **33.99%** |
> | FedAvg | $\alpha$-WFAT: $\frac{1+\alpha}{1-\alpha}=1.2$ |  emphasize   |   62.26%    |   44.08%   |   35.83%   |   33.31%   |
> | FedAvg |      FAT: $\frac{1+\alpha}{1-\alpha}=1.0$      |   original   |   57.45%    |   39.44%   |   32.58%   |   30.52%   |
> | FedAvg | $\alpha$-WFAT: $\frac{1+\alpha}{1-\alpha}=0.8$ | de-emphasize |   50.45%    |   34.34%   |   27.86%   |   26.62%   |
> | FedAvg | $\alpha$-WFAT: $\frac{1+\alpha}{1-\alpha}=0.6$ | de-emphasize |   40.47%    |   28.91%   |   24.36%   |   23.19%   |
> |        |               **SVHN (Non-IID)**               |              | **Natural** |  **FGSM**  | **PGD-20** |   **CW**   |
> | FedAvg | $\alpha$-WFAT: $\frac{1+\alpha}{1-\alpha}=1.4$ |  emphasize   |   90.60%    |   87.75%   | **73.12%** | **70.51%** |
> | FedAvg | $\alpha$-WFAT: $\frac{1+\alpha}{1-\alpha}=1.2$ |  emphasize   | **91.25%**  | **88.28%** |   71.72%   |   69.79%   |
> | FedAvg |      FAT: $\frac{1+\alpha}{1-\alpha}=1.0$      |   original   |   91.24%    |   87.95%   |   68.87%   |   67.89%   |
> | FedAvg | $\alpha$-WFAT: $\frac{1+\alpha}{1-\alpha}=0.8$ | de-emphasize |   90.03%    |   86.12%   |   64.35%   |   64.32%   |
> | FedAvg | $\alpha$-WFAT: $\frac{1+\alpha}{1-\alpha}=0.6$ | de-emphasize |   89.46%    |   84.80%   |   58.64%   |   58.96%   |
>
> [1] Hitaj, Dorjan, et al. "Evaluating the Robustness of Geometry-Aware Instance-Reweighted Adversarial Training." In arXiv. 2021.
>
> > Q4: Several notations are unclear, for example L-smoothness: l(.,.) is L-smoothness. Do you mean l(f\_theta(.),.) is L-smooth w.r.t. theta uniformly for all tilde x and y? Thm 4.2: $E|\nabla L^\alpha(\hat K)-\nabla_\theta(f(\tilde x),y)|^2$, what's the tilde x and y, one random sample? Thm 4.3: $E|\theta^0-\theta^*|^2$ term. What is the meaning of this expectation?
>
> **A4:** Thank you for the question. l(.,.) is L-smoothness indicates that the loss function satisfies the Lipschitzian smoothness condition w.r.t. model parameter $\theta$ for the adversarial training example $\tilde{x}$. For Theorem 4.2, $\tilde{x}$ means the adversarial variants of x w.r.t. y. For Theorem 4.3, the term means the expectation of difference between the initial model parameters with the optimal model parameters.

---

> ### Author Response · Authors · 2021-11-22
> **Need further clarification?**
>
> Thanks very much for your constructive comments on our work. We have tried our best to address the concerns. Is there any unclear point so that we should/could further clarify?

---

> ### Author Response · Authors · 2021-11-26
> **Would you mind confirming if you have further questions? Thanks!**
>
> Dear Reviewer 4ioa,
>
> As approach the end of the discussion period, we highly appreciate knowing if our responses have addressed your initial questions. We are delighted to answer your remaining concern. We appreciate your inputs and feedback very much. Thank you!
>
> Best wishes,
> The authors of Paper2033

---

### Official Review · Reviewer_4Hkz · 2021-11-02

**Correctness:** 3
**Technical Novelty And Significance:** 2
**Empirical Novelty And Significance:** 2
**Recommendation:** 5
**Confidence:** 5

**Main Review:**

Strengths:
(1) It relaxed the inner-maximization of Federated Adversarial Training by emphasizing the clients which could produce the larger margin for the decision boundary.
(2) It proposed a novel \alpha-Weighted Federated Adversarial Training approach, and provided its convergence in FL setting.
(3) The experiments showed that the proposed approach outperformed baseline FAT.

Weaknesses:
(1) The motivation of the relaxation on Federated Adversarial Training is counterintuitive. When \alpha-WFAT minimizes the lower bound of the FAT loss, it achieves much better performance than the original FAT. It implies the loss of FAT might include some reductant terms in the context of federated learning. In addition, it needs more discussion regarding the rationality of FAT in the non-IID FL, as federated learning is originally designed for IID centralized setting.
(2) The \alpha-weighted mechanism of Eq. (3) is not well motivated. Following the motivation in section 4.1, it simply emphasizes the client with the smaller loss. This is more directly correlated with the discrimination of the adversarial examples, instead of the margin that measures the uncertainty of predicting the adversarial examples.
(3) The experiment indicated that the hyper-parameter \alpha could significantly affect the model performance. But the selection or estimation of \alpha in the proposed FL method is not clear.
(4) The non-IID results are sensitive to the selection of data in each client. Thus it is better to report the results (mean and standard deviation) after running the experiments for multiple times.
(5) Some related works of adversarially robust federated learning are not discussed. For examples, some of them pointed out that local clients have limited computational resources to afford the local adversarial training. In this case, how would \alpha-WFAT determine the weight when some of them could only upload the standard training parameters?
[ref 1] Shah, Devansh, Parijat Dube, Supriyo Chakraborty, and Ashish Verma. "Adversarial training in communication constrained federated learning." arXiv preprint arXiv:2103.01319 (2021).
[ref 2] Reisizadeh, Amirhossein, Farzan Farnia, Ramtin Pedarsani, and Ali Jadbabaie. "Robust Federated Learning: The Case of
Affine Distribution Shifts." In NeurIPS. 2020.

Some other concerns:
(1) The \Tilde(x) in Theorem 4.2 is not defined. In this theorem, what would the index “t” and “k” start from?


**Summary Of The Paper:**

This work studied the limitation of conventional Federated Adversarial Training approach, and proposed an \alpha-weighted relaxation for Adversarial Training in the federated learning setting. Then it proposed a novel \alpha-Weighted Federated Adversarial Training for minimizing a lower bound of the inner-maximization in Federated Adversarial Training. The performance of the proposed \alpha-Weighted Federated Adversarial Training were validated for both IID and Non-IID federated learning settings.

**Summary Of The Review:**

The proposed techniques are not well motivated. The rationality of the proposed method on non-IID FL setting needs more explanation.

---

> ### Author Response · Authors · 2021-11-21
> **Response to Reviewer 4Hkz: Part1**
>
> > Q1 The motivation of the relaxation on Federated Adversarial Training is counterintuitive. When $\alpha$-WFAT minimizes the lower bound of the FAT loss, it achieves much better performance than the original FAT. It implies the loss of FAT might include some reductant terms in the context of federated learning. In addition, it needs more discussion regarding the rationality of FAT in the non-IID FL, as federated learning is originally designed for IID centralized setting.
>
> **A1:** It is actually the bias-variance trade-off [1]. The inner-maximization exacerbates the heterogeneity issue, yielding the slow convergence or even falling into a local optimal. We introduce the small bias to the original objective via a low-bound relaxation, facilitating a friendly optimization in the combination of Adversarial Training and Federated Learning. Theorem 4.2 points out such a trade-off in the objective relaxation by analyzing the convergence property. Besides, this relaxation is orthogonal to and compatible with the optimization methods like FedProx and Scaffold as demonstrated in Table 2. We will add more discussion about this bias-variance trade-off in the introduction to improve the clarity.
>
> Regarding the rationality of FAT in the non-IID FL, we will add more discussion in the submission to analyze the relationship between the existing optimization methods and the objective relaxation in FAT.
>
> [1] Kohavi, Ron, and David H. Wolpert. "Bias plus variance decomposition for zero-one loss functions." In ICML. 1996.
>
> > Q2: The $\alpha$-weighted mechanism of Eq. (3) is not well motivated. Following the motivation in section 4.1, it simply emphasizes the client with the smaller loss. This is more directly correlated with the discrimination of the adversarial examples, instead of the margin that measures the uncertainty of predicting the adversarial examples.
>
> **A2:** Thanks for questioning the point. We meant to use the distance between the correctly classified adversarial examples and the decision boundary (*i.e.,* the bold line) in Figure 2 to reflect the client loss, and told that $\ell_{client A}\>\ell_{client B}$. Then, selectively treating two client models in the aggregation can acquire a better global model. This is consistent with the intuition of Eq. (3). However, we admit that the unmarked distance might confuse the understanding and indirectly connect with Eq.(3). We have improved this unclear point by marking the distance in Figure 2 and added more descriptions about their connections in the caption and near Eq.(3). Please see the updated submission.
>
> > Q3: The experiment indicated that the hyper-parameter $\alpha$ could significantly affect the model performance. But the selection or estimation of $\alpha$ in the proposed FL method is not clear.
>
> **A3:** The choice of the hyper-parameter $\alpha$ in the lower bound scales up the different bias to the original objective. One useful way to set it is progressively probing its effect in a value-growth manner. When the $\alpha$ is very small, the objective will approximately degenerate the original objective of FAT, so does the performance with no harm. Slightly enlarging $\alpha$ can improve the performance due to the benefit of the bias-variance trade-off, and then make a stop at one point where the performance becomes drop. We will add the discussion of this empirically tweaking experience in the submission.

---

> ### Author Response · Authors · 2021-11-21
> **Response to Reviewer 4Hkz: Part2**
>
> > Q4: The non-IID results are sensitive to the selection of data in each client. Thus it is better to report the results (mean and standard deviation) after running the experiments for multiple times.
>
> **A4:** Thank you for your suggestion. We have run the multiple times of experiments and the newest results are as follows (in Table S1) and updated in the submission. In summary, the results confirmed that our $\alpha$-WFAT can consistently improve the model robustness with comparable or even better natural performance on the Non-IID settings.
>
> **Table S1.** Performance on **Non-IID** settings under different federated optimization methods (Mean$\pm$Std).
>
> | **CIFAR-10**  |                   |     **Natural**     |      **FGSM**       |     **PGD-20**      |       **CW**        |       **AA**        |
> | :-----------: | :---------------: | :-----------------: | :-----------------: | :-----------------: | :-----------------: | :-----------------: |
> |    FedAvg     |        FAT        |   58.13$\pm$0.68%   |   40.06$\pm$0.62%   |   32.56$\pm$0.01%   |   30.88$\pm$0.37%   |   29.17$\pm$0.03%   |
> |    FedAvg     | **$\alpha$-WFAT** | **63.36$\pm$0.07%** | **44.82$\pm$0.32%** | **37.14$\pm$0.03%** | **33.39$\pm$0.61%** | **31.66$\pm$0.70%** |
> |    FedProx    |        FAT        |   59.95$\pm$0.45%   |   41.44$\pm$0.15%   |   33.83$\pm$0.01%   |   31.65$\pm$0.36%   |   30.11$\pm$0.09%   |
> |    FedProx    | **$\alpha$-WFAT** | **62.04$\pm$0.47%** | **44.21$\pm$0.08%** | **36.64$\pm$0.11%** | **32.62$\pm$0.20%** | **31.83$\pm$0.15%** |
> |   Scaffold    |        FAT        |   61.44$\pm$1.37%   |   42.85$\pm$0.76%   |   34.08$\pm$0.05%   |   32.56$\pm$0.02%   |   31.03$\pm$0.08%   |
> |   Scaffold    | **$\alpha$-WFAT** | **63.16$\pm$0.96%** | **45.55$\pm$0.50%** | **37.33$\pm$0.02%** | **34.82$\pm$0.04%** | **33.32$\pm$0.01%** |
> | **CIFAR-100** |                   |     **Natural**     |      **FGSM**       |     **PGD-20**      |       **CW**        |       **AA**        |
> |    FedAvg     |        FAT        |   31.93$\pm$0.85%   |   20.04$\pm$0.44%   |   16.34$\pm$0.18%   |   13.65$\pm$0.28%   |   12.70$\pm$0.03%   |
> |    FedAvg     | **$\alpha$-WFAT** | **34.80$\pm$0.37%** | **20.91$\pm$0.35%** | **16.66$\pm$0.06%** | **13.78$\pm$0.13%** | **12.79$\pm$0.04%** |
> |    FedProx    |        FAT        | **34.07$\pm$0.74%** |   20.49$\pm$0.29%   |   16.20$\pm$0.12%   |   13.68$\pm$0.06%   |   12.67$\pm$0.08%   |
> |    FedProx    | **$\alpha$-WFAT** |   33.95$\pm$0.39%   | **20.86$\pm$0.40%** | **16.73$\pm$0.01%** | **13.80$\pm$0.04%** | **12.80$\pm$0.02%** |
> |   Scaffold    |        FAT        | **39.89$\pm$1.28%** |   24.78$\pm$0.40%   |   19.82$\pm$0.20%   |   16.73$\pm$0.01%   |   15.51$\pm$0.02%   |
> |   Scaffold    | **$\alpha$-WFAT** |   39.80$\pm$1.27%   | **25.05$\pm$0.51%** | **20.27$\pm$0.13%** | **16.79$\pm$0.01%** | **15.58$\pm$0.01%** |
> |   **SVHN**    |                   |     **Natural**     |      **FGSM**       |     **PGD-20**      |       **CW**        |       **AA**        |
> |    FedAvg     |        FAT        | **91.52$\pm$0.28%** |   88.13$\pm$0.18%   |   68.98$\pm$0.11%   |   68.04$\pm$0.15%   |   66.59$\pm$0.04%   |
> |    FedAvg     | **$\alpha$-WFAT** |   91.26$\pm$0.01%   | **88.27$\pm$0.02%** | **72.04$\pm$0.32%** | **69.96$\pm$0.16%** | **68.89$\pm$0.27%** |
> |    FedProx    |        FAT        |   91.00$\pm$0.08%   |   87.65$\pm$0.15%   |   68.48$\pm$0.04%   |   67.16$\pm$0.02%   |   65.76$\pm$0.18%   |
> |    FedProx    | **$\alpha$-WFAT** | **91.19$\pm$0.06%** | **88.15$\pm$0.01%** | **71.84$\pm$0.30%** | **69.88$\pm$0.35%** | **68.84$\pm$0.37%** |
> |   Scaffold    |        FAT        |   90.82$\pm$0.87%   |   87.89$\pm$0.66%   |   69.51$\pm$0.84%   |   68.12$\pm$0.88%   |   67.19$\pm$0.54%   |
> |   Scaffold    | **$\alpha$-WFAT** | **90.93$\pm$0.76%** | **88.27$\pm$0.45%** | **71.77$\pm$0.38%** | **69.49$\pm$0.67%** | **68.37$\pm$0.48%** |

---

> ### Author Response · Authors · 2021-11-21
> **Response to Reviewer 4Hkz: Part3**
>
> > Q5: Some related works of adversarially robust federated learning are not discussed. For examples, some of them pointed out that local clients have limited computational resources to afford the local adversarial training. In this case, how would $\alpha$-WFAT determine the weight when some of them could only upload the standard training parameters? [ref 1] Shah, Devansh, Parijat Dube, Supriyo Chakraborty, and Ashish Verma. "Adversarial training in communication constrained federated learning." arXiv preprint arXiv:2103.01319 (2021). [ref 2] Reisizadeh, Amirhossein, Farzan Farnia, Ramtin Pedarsani, and Ali Jadbabaie. "Robust Federated Learning: The Case of Affine Distribution Shifts." In NeurIPS. 2020.
>
> **A5:** Actually, we have discussed [ref 1] about the transmission budget in the submission. We will definitely include the related [ref 2] when discussing the robust learning for the heterogeneity issues in the federated learning.
>
> > Q6: Some other concerns: (1) The $\tilde{x}$ in Theorem 4.2 is not defined. In this theorem, what would the index “t” and “k” start from?
>
> **A6:** Thank you. We have corrected the notation typos and clearly define them in the updated version, and the index "t" and "k" start from 1.

---

> ### Author Response · Authors · 2021-11-22
> **Need further clarification?**
>
> Thanks very much for your constructive comments on our work. We have tried our best to address the concerns. Is there any unclear point so that we should/could further clarify?

---

> ### Author Response · Authors · 2021-11-26
> **Would you mind confirming if you have further questions? Thanks!**
>
> Dear Reviewer 4Hkz,
>
> As approach the end of the discussion period, we highly appreciate knowing if our responses have addressed your initial questions. We are delighted to answer your remaining concern. We appreciate your inputs and feedback very much. Thank you!
>
> Best wishes,
> The authors of Paper2033

---

> > ### Comment · Reviewer_4Hkz · 2021-11-28
> > **Thank you for your response!**
> >
> > I would like to keep my score unchanged for the following reasons.
> > (1) The bias-variance explanation of the proposed low-bound relaxation is not clear. How are the bias and variance defined in this case? Why would introducing such a bias improve the model performance from the perspective of bias-variance trade-off?
> > (2) In the experiments, how would the distance between the correctly classified adversarial examples and the decision boundary be estimated? Can cross-entropy loss on those adversarial examples be applied to determine such a distance? In Lines 7-8 of Algorithm 1, the adversarial loss will be ranked by N_k/N L_k = 1/N \sum_{i=1}^{N_k} l(x_i). That is, the clients with more adversarial examples are more likely to have larger adversarial loss due to the accumulated computation. From this point of view, would clients with more data be more likely to be emphasized?
> > (3) The value-growth method in learning the hyper-parameter \alpha assumes that the model performance is convex with respect to \alpha. That is because it stops at one point where the performance becomes drop. It is unclear how this can be explained by bias-variance trade-off, especially when it is observed [1-2] that bias-variance trade-off of deep neural networks is complicated.
> > [1] Belkin, Mikhail, Daniel Hsu, Siyuan Ma, and Soumik Mandal. "Reconciling modern machine-learning practice and the classical bias–variance trade-off." Proceedings of the National Academy of Sciences 116, no. 32 (2019): 15849-15854.
> > [2] Yang, Zitong, Yaodong Yu, Chong You, Jacob Steinhardt, and Yi Ma. "Rethinking bias-variance trade-off for generalization of neural networks." In International Conference on Machine Learning, pp. 10767-10777. PMLR, 2020.

---

> > > ### Author Response · Authors · 2021-11-28
> > > **Thank you for the valuable questions!**
> > >
> > > > Q1: The bias-variance explanation of the proposed low-bound relaxation is not clear. How are the bias and variance defined in this case? Why would introducing such a bias improve the model performance from the perspective of bias-variance trade-off?
> > >
> > > **A1:** In terms of the combination of adversarial training and federated learning, the key point is that such adversarial examples exacerbate the heterogeneity via inner-maximization, which induces performance deterioration under the model aggregation methods. In Figure 2, we roughly illustrate the radical optimization caused by client local data (e.g., in ClientA), therefore we introduce a biased objective relaxation to facilitate a friendly optimization. We wish this explanation can give you an intuitive understanding of the reason why such bias can improve the model performance.
> > >
> > > > Q2: In the experiments, how would the distance between the correctly classified adversarial examples and the decision boundary be estimated? Can cross-entropy loss on those adversarial examples be applied to determine such a distance? In Lines 7-8 of Algorithm 1, the adversarial loss will be ranked by N_k/N L_k = 1/N \sum_{i=1}^{N_k} l(x_i). That is, the clients with more adversarial examples are more likely to have larger adversarial loss due to the accumulated computation. From this point of view, would clients with more data be more likely to be emphasized?
> > >
> > > **A2:** In the experiments, we utilize the cross-entropy loss on those adversarial examples to decide which clients should be emphasized, it is based on our introduced objective relaxation for federated adversarial training. We do not claim that we can accurately estimate the distance between correctly classified adversarial examples and the decision boundary. Our main design is introducing the relaxation in the original objective to avoid the radical optimization of some clients on local data, and we use Figure 2 to illustrate the radical optimization (e.g., in ClientA) and our high-level idea (emphasizing the client like ClientB). As for the assumption that radical optimization induces the performance drop, we also conduct the experiment (in A3&Q3 for Reviewer 4ioa or in Appendix E.1) that de-emphasizes the client with smaller adv loss (relatively emphasize other clients with larger adv loss) results in worse performance.  In our experiments, we actually keep all clients with the same local training data, so the current results are not attributed to emphasizing clients with more data.  In the detailed implementation, we calculate the averaged adversarial loss value on each client's local data to rank those clients.
> > >
> > > > Q3: The value-growth method in learning the hyper-parameter \alpha assumes that the model performance is convex with respect to \alpha. That is because it stops at one point where the performance becomes drop. It is unclear how this can be explained by bias-variance trade-off, especially when it is observed [1-2] that the bias-variance trade-off of deep neural networks is complicated.
> > > >
> > > > [1] Belkin, Mikhail, Daniel Hsu, Siyuan Ma, and Soumik Mandal. "Reconciling modern machine-learning practice and the classical bias–variance trade-off." Proceedings of the National Academy of Sciences 116, no. 32 (2019): 15849-15854. [2] Yang, Zitong, Yaodong Yu, Chong You, Jacob Steinhardt, and Yi Ma. "Rethinking bias-variance trade-off for generalization of neural networks." In International Conference on Machine Learning, pp. 10767-10777. PMLR, 2020.
> > >
> > > **A3:** We have conducted the empirical study about varying the $\alpha$ in the third panel of Figure 4. The results indicate that an appropriate $\alpha$ (e.g., $\alpha$>0) can indeed improve the model performance but the larger $\alpha$ may induce the performance drop which can be attributed to the over-emphasized model aggregation according to our objective relaxation (i.e., Eq.(3)).  It can be explained by our Eq.(3) that enlarging $\alpha$ will emphasize the client with the smaller adversarial loss which can avoid radical optimization by those clients with larger adversarial loss, but a larger $\alpha$ can cause the over-emphasize on some specific data. For example, consider an extreme case, the overall objective will approximate to minimize the adversarial loss in one specific client and that is unreasonable and will undermine its generalization on other clients' data. As for the bias-variance trade-off, we will cite these papers to have a discussion.
> > >
> > >
> > >
> > > We highly appreciate knowing whether our response can address your questions and are delighted to have a further discussion if you have any remaining concerns!

---

### Official Review · Reviewer_1hs3 · 2021-11-03

**Correctness:** 4
**Technical Novelty And Significance:** 4
**Empirical Novelty And Significance:** 4
**Recommendation:** 8
**Confidence:** 3

**Main Review:**

Pros:

1. This paper is the first to solve the robustness-drop problem in federated adversarial training. It has great practical impacts to real-world applications.

2. The authors provided an intuitive assumption on the cause of robustness-drop problem, which is further verified by the experiments.

3. The proposed \alpha-weighted federated adversarial training method is technically solid, easy to implement, and empirically achieves good performance under multiple different adversarial training and federated learning settings.

4. Ablation studies show the proposed model is robust to hyper-parameter choices.

Suggestions:

I don't see any obvious cons of this paper. One piece of suggestion is that, if your assumption on the cause of the robustness-drop phenomenon is correct, maybe combining "benign adversarial examples" [1] with \alpha-WFAT can even further improve performance. The intuition is that "benign adversarial examples" may not amplify the distribution heterogeneously so much as the traditional harsh adversarial samples.

[1] Attacks Which Do Not Kill Training Make Adversarial Learning Stronger.


**Summary Of The Paper:**

This paper tackles an important problem in federated adversarial training: robustness accuracy significantly drops at the later stage of training. The authors first raise their assumption for the cause of this phenomenon: Adversarial training amplifies the heterogeneity of data distributions across different clients, and overfitted local robustness can not well generalized to other clients. Based on this assumption, the authors proposed \alpha-weighted federated adversarial training, which essentially up-weights the model trained on benign distributions and down-weights those on harsh distributions when averaging them up at the cloud center. Results show the proposed method outperforms previous state-of-the-arts under different adversarial training and federated learning settings.

**Summary Of The Review:**

A solid paper providing the first solution for an important problem in federated adversarial training.

---

> ### Author Response · Authors · 2021-11-21
> **Response to Reviewer 1hs3**
>
> > Q1:  I don't see any obvious cons of this paper. One piece of suggestion is that, if your assumption on the cause of the robustness-drop phenomenon is correct, maybe combining "benign adversarial examples" [1] with $\alpha$-WFAT can even further improve performance. The intuition is that "benign adversarial examples" may not amplify the distribution heterogeneously so much as the traditional harsh adversarial samples.
>
> > [1] Attacks Which Do Not Kill Training Make Adversarial Learning Stronger.
>
> **A1:** Thank you for the suggestion! In this paper, $\alpha$-WFAT mainly leverages the client-level measure to alleviate the heterogeneous issue in federated adversarial training. It is compatible and useful to further incorporate the fine-grained instance-level measure [1] to improve the performance. We will specially discuss the possible extension by combining $\alpha$-WFAT and "benign adversarial examples" [1] in our paper.

---

### Decision · Program_Chairs · 2022-01-20

**Decision:**

Reject

**Comment:**

This manuscript proposes and analyses a weighting approach to improve the conformance of adversarial training in federated learning. The authors observe that adversarial training seems to degrade during the late stages of training, and suggest that this degradation is a consequence of exacerbated cross-device bias in federated averaging. They suggest and analyze a weighted scheme to fix this issue.

During the review, the main concerns are related to the novelty of the work compared to existing work, the clarity of the technical contributions, and unclear technical statements. The authors respond to these concerns and partially satisfy the reviewers. After discussion, reviewers remain mixed, with multiple weak rejects and one strong accept. No fatal flaws are noted.

The opinion of the area chair is that while there are no fatal flaws, there is very limited enthusiasm for this paper. This limited enthusiasm seems to be a result of intuition for observed phenomena that seem incorrect or insufficient to reviewers. Overall, I think this paper outlines and addresses an interesting issue of real concern. Flaws in the intuition building/explanation, and issues with clarity of presentation need to be improved for this work to have some impact.